# Lineage-specific microbial protein prediction enables large-scale exploration of protein ecology within the human gut

Matthias A. Schmitz ®[1], Nicholas J. Dimonaco ®[2,3], Thomas Clavel ®[1] & Thomas C. A. Hitch ®[1]✉

Microbes use a range of genetic codes and gene structures, yet these are often ignored during metagenomic analysis. This causes spurious protein predictions, preventing functional assignment which limits our understanding of ecosystems. To resolve this, we developed a lineage-specific gene prediction approach that uses the correct genetic code based on the taxonomic assignment of genetic fragments, removes incomplete protein predictions, and optimises prediction of small proteins. Applied to 9634 metagenomes and 3594 genomes from the human gut, this approach increased the landscape of captured expressed microbial proteins by 78.9%, including previously hidden functional groups. Optimised small protein prediction captured 3,772,658 small protein clusters, which form an improved microbial protein catalogue of the human gut (MiProGut). To enable the ecological study of a protein's prevalence and association with host parameters, we developed InvestiGUT, a tool which integrates both the protein sequences and sample metadata. Accurate prediction of proteins is critical to providing a functional understanding of microbiomes, enhancing our ability to study interactions between microbes and hosts.

Microbiome analysis has most often focused on the study of taxonomic groups, although it is the functionality of these taxa that is of interest[1,2]. Microbes that perform the same function, or work together to form a functional unit, are referred to as members of the same 'guild'[3]. Guild-based analysis of metagenomic datasets provides some functional insight but represents an indirect method of studying functionality, as the unit of study remains the taxa rather than directly studying the function of interest. Such inference of functionality from taxonomy is reductionist, given that even strains of the same species vary in their functionality, thereby expanding the functional capacity of microbial communities[4].

Methods derived from functional ecology[4–9] can be applied to the study of proteins and their functions. We term this 'protein ecology', which aims to study the ecological distribution of proteins or functions as the unit of study[2,6,10]. The direct study of proteins/functions takes into account horizontal gene transfer, which can lead to the sharing of the protein of interest outside the taxa being studied, although transcriptional activity and genomic context can modify the functionality of a protein[11–13]. It has been shown that horizontal gene transfer occurs at high rates in the human gut microbiome, particularly in industrialised populations[14], facilitating the spread of antibiotic resistance genes, virulence factors, and other traits that affect the progression of human disease[15].

Computational methods such as AnnoTree have been developed to study the distribution of functions across the microbial tree of life, highlighting functional variation between taxa[16]. Studying the ecology of proteins, rather than species, can provide insights into the ecological importance of protein-bearing species and the ecological pressures that drive protein evolution[10]. In the context of the human gut microbiome, species-independent functional studies have provided

[1]Functional Microbiome Research Group, RWTH University Hospital, Aachen, Germany. [2]Institute for Global Food Security, School of Biological Sciences, Queen's University Belfast, Belfast, UK. [3]Department of Computer Science, Aberystwyth University, Aberystwyth, UK. ✉e-mail: thitch@ukaachen.de

insights into the association of bile acid and sulphur metabolic genes with colorectal cancer[17,18], and type 3 secretion system effectors with human health conditions[19].

Gene catalogues also place the proteins/genes as the unit of study by providing a reference source of non-redundant genes for either descriptive studies of a microbiome[20], or targeted analysis of microbial functionality[21,22]. So far, each gene catalogue has ignored the diversity of genetic codes used by bacteria[23–25] and fails to account for the existence of eukaryotic multiple exon genes[26]. This is indicative of the trend in human gut microbiota research to focus on prokaryotes, neglecting other domains[27]. This is detrimental as eukaryotes and viruses directly influence human health, including via immune modulation[28].

Further to this, several critical issues in genome annotation, particularly when applied across diverse taxa have been identified[29]. One of the major challenges identified is the variability in gene prediction accuracy, where certain tools excel at predicting specific gene types in particular taxa, but fail when applied to others. This inconsistency stems from the vast diversity in genetic structures, including variations in coding sequences, regulatory elements, and the genetic codes used by different organisms. Additionally, prokaryotic annotation tools often perform poorly when applied to eukaryotic genes, especially those with complex exon-intron structures, while tools designed for eukaryotic genomes may overlook small, overlapping genes commonly found in prokaryotes[30]. The limitations of current annotation tools are compounded by the lack of comprehensive training datasets, particularly for non-model organisms, especially lacking those from diverse lineages, leading to errors in gene predictions and functional annotations. As a result, the incomplete or inaccurate annotation of metagenomic datasets can obscure important biological functions, particularly in diverse microbial communities. These challenges have started to be addressed by the introduction of 're-annotation' techniques[31] and initiatives to clean-up consensus assemblies and annotations[32,33]. While these solutions address genomes, microbiota vary greatly, meaning de novo gene prediction is essential. As of yet, no lineage-directed gene prediction method exists for microbiome analysis.

We present a workflow using the taxonomy of metagenomic contigs to inform protein sequence prediction. Application of this approach to the human gut uncovered a multitude of previously missed proteins and provided an improved gene catalogue. To facilitate protein ecology studies of the human gut, we developed InvestiGUT, a tool that identifies associations between protein sequence prevalence and host parameters.

## Results

### Lineage-specific gene prediction expands the human gut protein repertoire

A workflow for lineage-optimised gene prediction was developed by selecting gene prediction tools based on the taxonomic assignment of each metagenomically assembled sequence and customisation of each tool's parameters (genetic code, gene size) (Fig. 1a). Tool selection for gene prediction was informed by initial testing of 13 tools on archaeal ($n = 3$), bacterial ($n = 17$), fungal ($n = 3$), and viral ($n = 3$) species (total species = 26). The quality of annotations from each tool was quantified and compared using ORForise[29]. Given the variability of an individual tool's ability to predict all genes, we investigated the potential synergy of combining two or three tools in tandem (Supplementary Table 3). Given the small, but consistent increase in both full and partial genes predicted by the combination of three tools, and the low cost in terms of spurious genes compared to two tools, we chose to use the combination of three tools that performed best for each taxonomic group. While this approach does result in an inflation of spurious predictions, below we have applied multiple approaches to study the benefit of this approach. This includes metatranscriptomic analysis to identify

evidence for expression, and comparison with an independent gene catalogue to provide independent validation. Based on these results, we believe it is more advantageous to predict additional real proteins at the expense of including spurious genes, than to risk excluding real proteins.

This workflow was applied to 9677 metagenomes from 28 countries (Fig. 1b). In addition to the metagenomes, a non-redundant collection of genomes representing the prokaryotic diversity within the human gut was included for downstream analysis of the taxonomic occurrence of proteins[34]. Taxonomic profiling of the metagenomic samples was consistent with previous observations that identified Bacteria as the dominant microbial domain in the human gut (Supplementary Table 5). The annotation of contigs was observed to be dependent on the database used, consistent with previous findings and highlighting the need for greater characterisation of novel taxa to improve coverage by the databases (Supplementary Table 6)[35]. The predicted proteins were dominated by those originating from bacterial contigs ($58.4 \pm 18.9\%$), followed by proteins on contigs that could not be assigned to a specific domain by Kraken 2, termed unknown ($41.2 \pm 18.8\%$), then viruses ($0.19 \pm 0.41\%$), archaea ($0.15 \pm 0.65\%$), and eukaryotes ($0.03 \pm 1.31\%$) (Supplementary Table 7). The high percentage of taxonomically unassigned proteins, the unknown group, is in line with current estimates that >50% of the gut microbiota has yet to be cultured, hence complete genomes, which were used for taxonomic assignment, do not exist for many gut microbes[36,37].

In total, 846,619,045 genes (metagenomes: 838,528,977, genomes: 8,090,068) were predicted, with the majority originating from contigs of bacteria or unknown assignment (Fig. 1c). In comparison, the exclusive use of Pyrodigal across all metagenomes identified 737,874,876 genes (metagenomes: 730,237,038, genomes: 7,637,838), meaning that the lineage-specific workflow identified an additional 108,744,169 genes (14.7%). While the use of multiple gene prediction tools might be expected to inflate the number of predicted genes, the majority of predictions were consistent across tools, except for eukaryotes. For eukaryotic contigs, AUGUSTUS[38] resulted in substantially lower gene prediction numbers, reducing the harmonious prediction of eukaryotic genes across all three tools (Fig. 1d). In addition to the lower number of genes predicted by AUGUSTUS, the overlap between SNAP and Pyrodigal also highlights inconsistency. This is likely due to the inability of Pyrodigal to predict multi-exon genes, whereas both SNAP and AUGUSTUS predict exons and introns, which is crucial for accurate gene prediction in eukaryotes[39].

### Enhanced coverage of the protein landscape from the human gut

Lineage-specific gene prediction yielded a larger number of proteins compared to any single approach. Therefore, we sought to confirm that these were real proteins and not spurious predictions. To facilitate comparison with a previously established protein catalogue, the Unified Human Gastrointestinal Protein (UHGP) catalogue[24], we dereplicated our >800 million proteins at 90% similarity to 29,232,514 protein clusters, increasing the human gut protein landscape by 210.2% compared to UHGP. We termed this protein catalogue the Microbial Protein Catalogue of the Human Gut (MiProGut).

Nearly 10,000 samples were used to generate MiProGut, but the rarefaction of MiProGut suggests that greater diversity has yet to be captured for all taxonomic groups (Fig. 2a). This was supported by analysis of 100 permutations where 2397 samples were needed to cover 50% of proteins in MiProGut, and for 90% coverage an average of 7824 samples were needed. Eukaryotic proteins were observed least frequently, with the majority of these proteins being identified within a few samples that were dominated by Eukaryotic contigs (Supplementary Table 5). The Western bias of included samples, as well as low number of samples from developing countries may explain the rarefaction observation of further diversity. The inclusion of additional

 

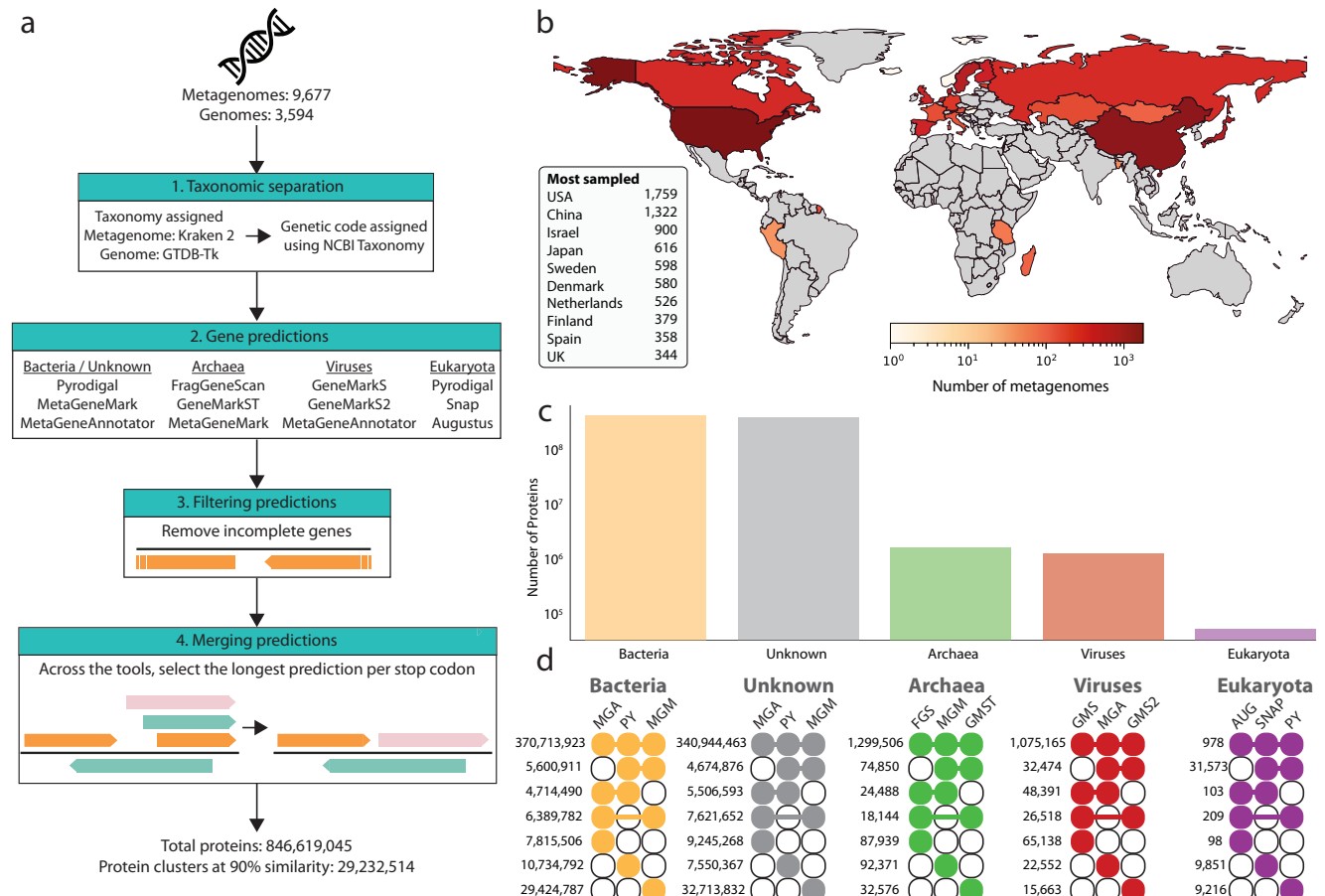

**Fig. 1 | Taxonomy-informed gene prediction workflow and its application to the human gut. a** The workflow consists of three major steps. Taxonomic separation of either metagenomic contigs, or input genomes, into their respective domains. The genetic code utilised by each lineage is then identified at the species level, to account for variation within domains of life. For each domain-genetic code grouping, the respective gene prediction tools are run, and then merged, removing redundant predictions and incomplete gene predictions occurring at the edges of contigs. **b** Geographical distribution of the human gut metagenomes used, with darker colours indicating a greater number of samples originate from the country. **c** The number of proteins predicted for each taxonomic domain. **d** The overlap between gene prediction tools was identified for each domain to quantify each tool's variation in predictions. Visualisation is based on a variation of an upset plot, where the number on the left is the number of overlapping protein predictions for each domain with the tools indicated by the connected coloured circles. Abbreviations for tools are: MGA MetaGeneAnnotator, MGM MetaGeneMark, FGS FragGeneScan, GMS GeneMarkS, GMS2 GeneMarkS2, GMST GeneMarkST, PY Pyrodigal, SNAP SNAP, AUG AUGUSTUS.

samples may also reduce the number of protein clusters consisting of a single protein sequence, known as singletons ($n = 14,043,436$). This frequency of singleton clusters is consistent with previous observations that most protein clusters are rare[25], leading to the formation of singletons. While singletons were rarely captured by metagenomic sequencing, metatranscriptomic expression in human gut samples was observed for 39.1% of singletons, confirming they are not spurious and are functionally relevant to the microbiota.

MiProGut provides an improved resource for protein sequence identification, but functional annotation is required to understand the role of these proteins, both in relation to the microbiome, and their impact on the host[40,41]. The majority (64.4%) of MiProGut proteins lacked informative functional annotation when annotated with eggNOG-mapper (Fig. 2b). Further annotation with MANTIS, which integrates multiple functional databases, increased the annotated fraction by 4.9% compared to all eggNOG assigned COG categories. However, this included assignments to general functions. These values re-highlight the high number of microbial proteins in the human gut that lack meaningful functional annotation[36].

Significant overlap was observed between MiProGut and UHGP, with 75.7% of UHGP proteins being covered by MiProGut, whereas only 37.6% of MiProGut proteins were covered by UHGP (Fig. 2c). The UHGP

proteins not captured by MiProGut are likely due to the different sources each catalogue used to predict proteins, namely the focus on genomes for UHGP and metagenomes for MiProGut. As UHGP was generated from prokaryotic genomes, the majority of MiProGut proteins matching UHGP were of bacterial, archaeal, or unknown origin. Conversely, only 16.6% of viral and 0.3% of eukaryotic MiProGut proteins matched UHGP due to their exclusion (Supplementary Table 8). Functionally this means MiProGut is enriched with eukaryotic functions, for example there were 413 protein clusters assigned to 'nuclear structure' in MiProGut while only seven in UHGP. While the identification of overlapping proteins between the two catalogues supports that these proteins have been observed previously, further evidence is required to validate them as real and not spurious.

Given that MiProGut accounted for a greater range of functionality, including that from non-bacterial domains, we aimed to quantify the impact of this on our ability to study the human gut microbiota. Therefore, we quantified the expression captured in 862 metatranscriptomic samples from three studies (Fig. 2d)[7,42,43]. We identified transcriptional evidence for 13,756,670 MiProGut proteins (47.1%), compared to 7,689,106 UHGP proteins (55.7%). This included 22.7% of the MiProGut protein clusters containing eukaryotic proteins, 35.9% of the viral-containing clusters, and 45.4% of archaeal-containing clusters.

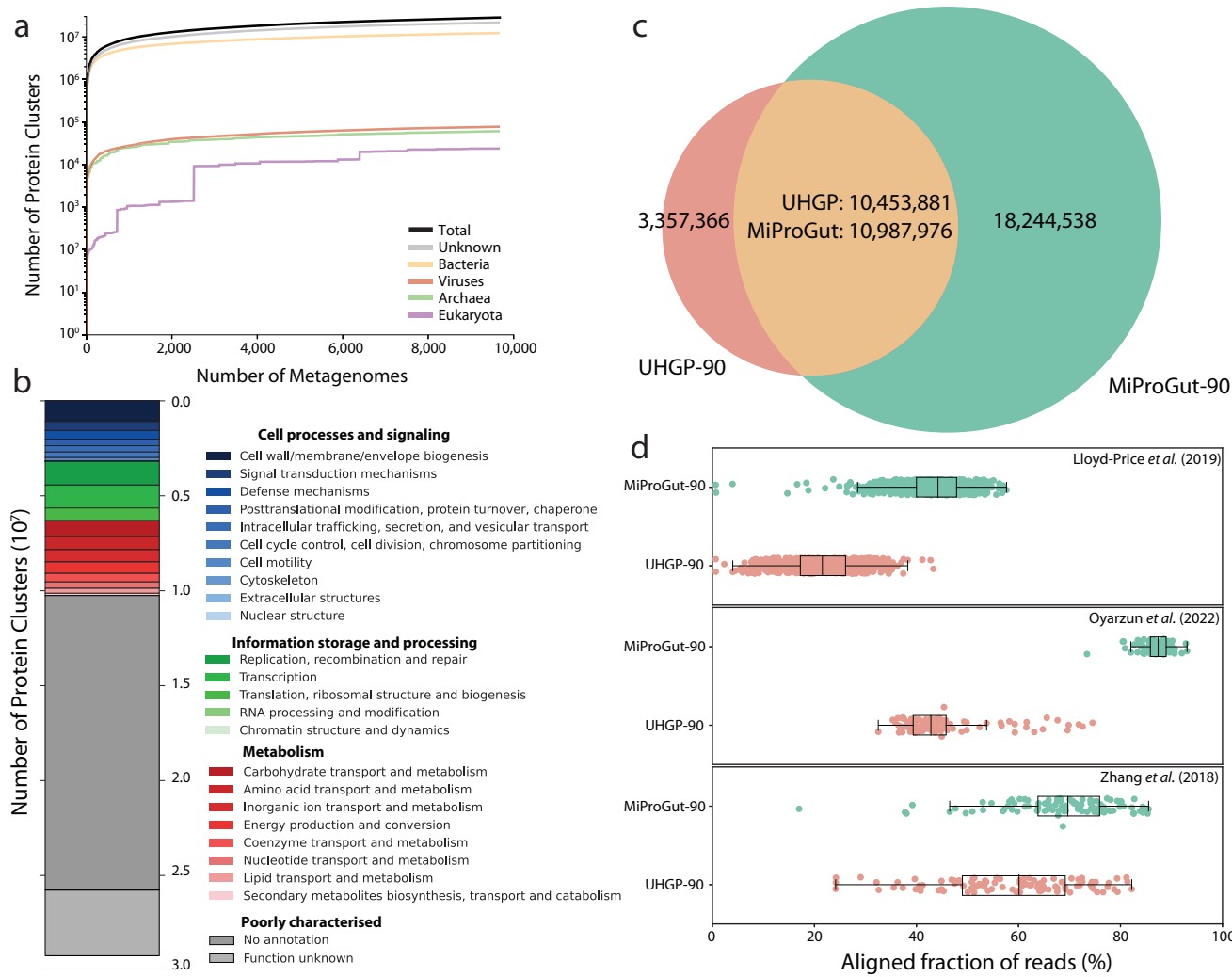

**Fig. 2 | Taxonomy-informed gene prediction expands the protein landscape of the human gut microbiome. a** For each taxonomic group, as well as the cumulative protein repertoire of the proteins within the MiProGut (log10 scale), rarefaction analysis was conducted across all studied metagenomes. **b** Functional annotation of the 29,232,514 MiProGut proteins based on COG functional groupings via eggNOG-mapper annotation. **c** Comparison of MiProGut (green) to UHGP (red), both clustered at 90% amino acid identity. **d** The aligned fraction of metatranscriptomic reads from multiple human gut studies (Lloyd-Price et al. 2019: $n = 682$, Oyarzun et al. 2022: $n = 100$, Zhang et al. 2018: $n = 100$) by both MiProGut and UHGP. The boxplots include a line in the centre indicating the median, the boxes represent the interquartile range, and the whiskers represent the minimum and maximum values, not including outliers.

Given the increased number of expressed proteins, we calculated the aligned fraction of reads for each of the 862 metatranscriptomes. Across all samples, MiProGut covered an additional 22.9 ± 9.9% reads compared to UHGP. The largest increase occurred in a Spanish cohort where MiProGut increased read coverage by an additional 41.6 ± 8.6%[42]. In addition to the expression of proteins, we used the prediction of proteins by multiple tools and the protein cluster, including multiple proteins (≥2 members), to create a high-quality subset of 25,266,245 (86.4%) protein clusters (MiProGut-HQ).

The inclusion of genomes representative of all prokaryotes detected in the human gut allowed us to investigate the occurrence of protein clusters across different phylogenomic distances. Of the 29,232,514 protein clusters in MiProGut, 19.6% (5,734,391) were detected within a genome, and only 927,384 (3.2%) occurred in genomes assigned to multiple species (Supplementary Fig 1a). To better assess the taxonomic range at which each protein cluster is shared, we evaluated the genus (Supplementary Fig 1b), and phylum level (Supplementary Fig 1c) and observed 142,342 and 19,320 protein clusters to be shared across multiple taxa at each level, respectively. To identify the most frequently shared protein clusters, we filtered for those

present in ≥10 species and ≥2 phyla. Of these 2423 clusters, 2197 (90.6%) were present in Firmicutes_A, a division of the phylum Bacillota (formerly Firmicutes) (Supplementary Fig 1d). As viruses are a known mediator of horizontal gene transfer within the gut[44] we investigated the protein clusters containing a protein of viral origin ($n = 77,496$) that also occurred in the reference genomes ($n = 2914$) (Supplementary Fig 1e). Of these, 75 protein clusters occurred in at least two phyla and were annotated as including virulence and antibiotic resistance genes (Supplementary Fig 1f). The most promiscuous viral protein cluster was identified as P4 primase83 (Supplementary Table 9), detected within 58 species, which may suggest that P4 phage acts as a corridor of horizontal gene transfer between core members of the gut. The horizontal transfer of genes from prokaryotes into fungi has been a driver of fungal evolution within gut ecosystems[45]. We identified 87 protein clusters occurring within the prokaryotic reference genomes, which also contain a eukaryotic protein, suggesting they may be examples of prokaryotic genes transferred to eukaryotes (Supplementary Fig 1g). These findings support previous findings that horizontal gene transfer frequently occurs between members of the human gut microbiome[14].

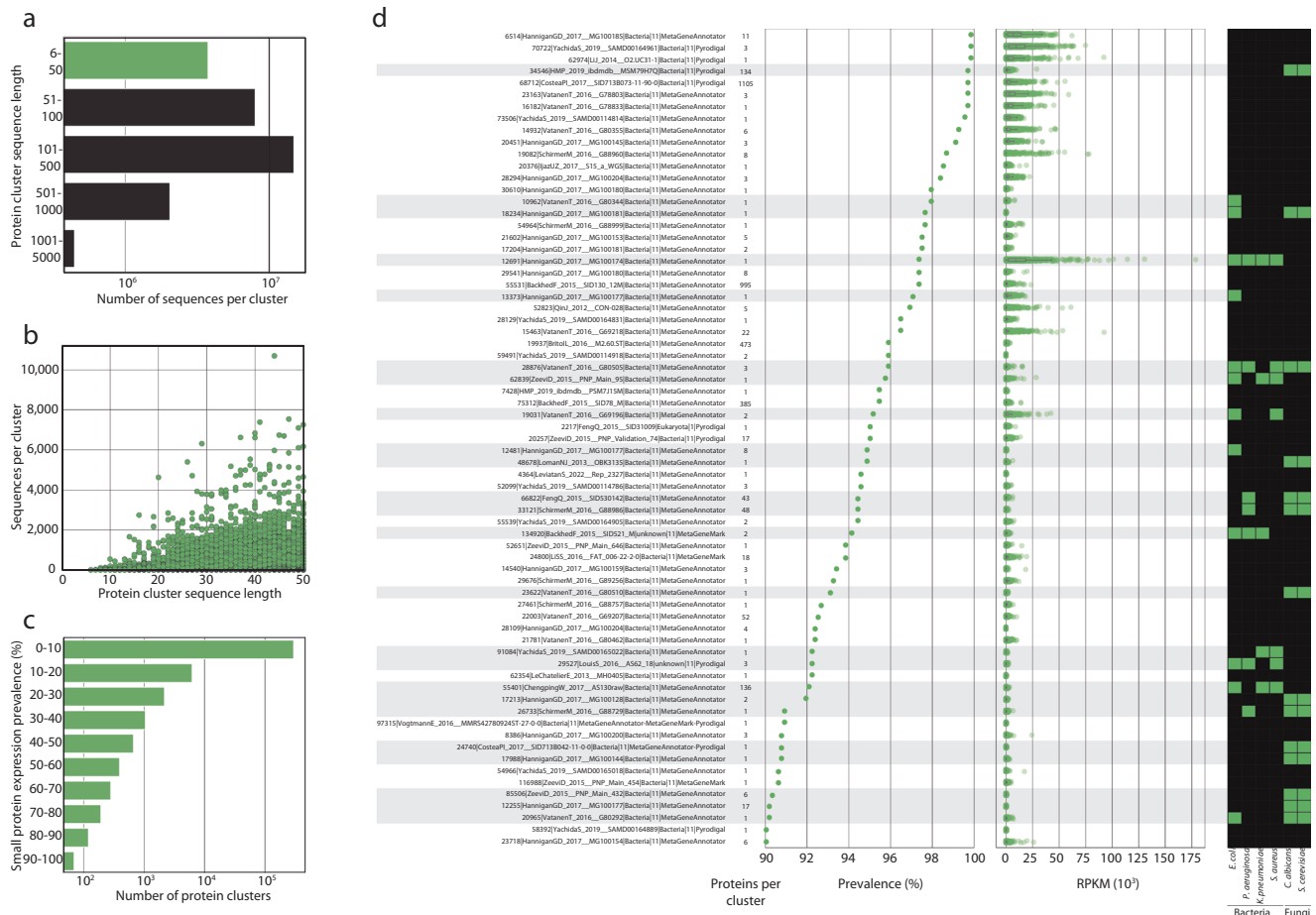

**Fig. 3 | Identification of highly prevalent, and expressed small proteins within the gut and their antimicrobial potential. a** The size of protein cluster representative sequences, with small proteins (≤50 amino acids) in green. **b** The number of members within each protein cluster, plotted against the size of each protein cluster's representative sequence. **c** The prevalence of each protein cluster's expression across the 687 metatranscriptomic samples from Lloyd-Price et al.[7].

**d** The name, number of member sequences, expression prevalence across the 687 metatranscriptomic samples from Lloyd-Price et al., RPKM of expression, and predicted antimicrobial activity for proteins expressed within ≥90% of metatranscriptomic samples (*n* = 69). The boxplots include a line in the centre indicating the median, the boxes represent the interquartile range, and the whiskers represent the minimum and maximum values, not including outliers.

## Identification of commonly expressed small protein clusters within the human gut

Small proteins have been understudied in the human gut as many gene prediction tools exclude them by default[46]. To improve the prediction of small proteins, the gene prediction parameters were modified to include proteins of >5 amino acids. This resulted in the prediction of 44,164,853 small proteins (5.2% of total proteins), represented by 3,571,095 protein clusters within MiProGut, called small protein clusters (SPCs) (Fig. 3a). Of these, 1,104,393 (30.9%) clusters were singletons (Fig. 3b), while the largest SPC contained 10,700 proteins and was identified as a member of the proposed 'family 350024' of crosstalk proteins[46]. Studying the expression of the SPCs within 687 metatranscriptomic samples revealed that all SPCs were expressed within at least one sample, but many were rarely expressed (<10% of samples). However, 69 SPCs were highly prevalent, being expressed in >90% of samples (Fig. 3c).

In addition to being largely ignored, small proteins contribute critical functionality to the survival of strains within complex communities. One such functionality is antimicrobial activity, of which the gastrointestinal tract has been identified as a rich source[47,48]. To determine whether this included the most commonly expressed SPCs, we predicted each SPCs antimicrobial activities[49] (Fig. 3d). By predicting the antimicrobial activity against four ESKAPE pathogens and two fungi of relevance to human health, we found that 35% (24/69) of

these SPCs could be antimicrobial. The most highly expressed antimicrobial SPC (13,707 ± 17,886 RPKM) was predicted to be active against all four ESKAPE pathogens (*Escherichia coli* ATCC 25922, *Pseudomonas aeruginosa* ATCC 27853, *Klebsiella pneumoniae, Staphylococcus aureus* ATCC 25923). Although expressed in 97.1% of the samples, this cluster contained only a single sequence with similarity to an unknown protein from *Bifidobacterium* spp. The presence of this protein within a dominant genus of human gut commensals, in addition to its high expression and wide range of predicted activity, suggests this protein may be of importance to the gut ecosystem. Of the 24 antimicrobial SPCs, 14 were predicted to target both fungal species, *Candida albicans* and *Saccharomyces cerevisiae*.

## Ecological distribution of proteins across individuals uncovers associations with host health status

The application of lineage-specific gene prediction enhances the ability to study the ecological distribution of proteins, referred to in this manuscript as protein ecology. Using the metadata associated with both the metagenomes studied, and the genomes analysed, we created a tool, InvestiGUT, to facilitate protein ecology studies of the human gut (Fig. 4).

InvestiGUT was developed to accept multiple sequences at once and examines them either individually or as a collection. If proteins are studied together as a collection, only samples that contain all queried

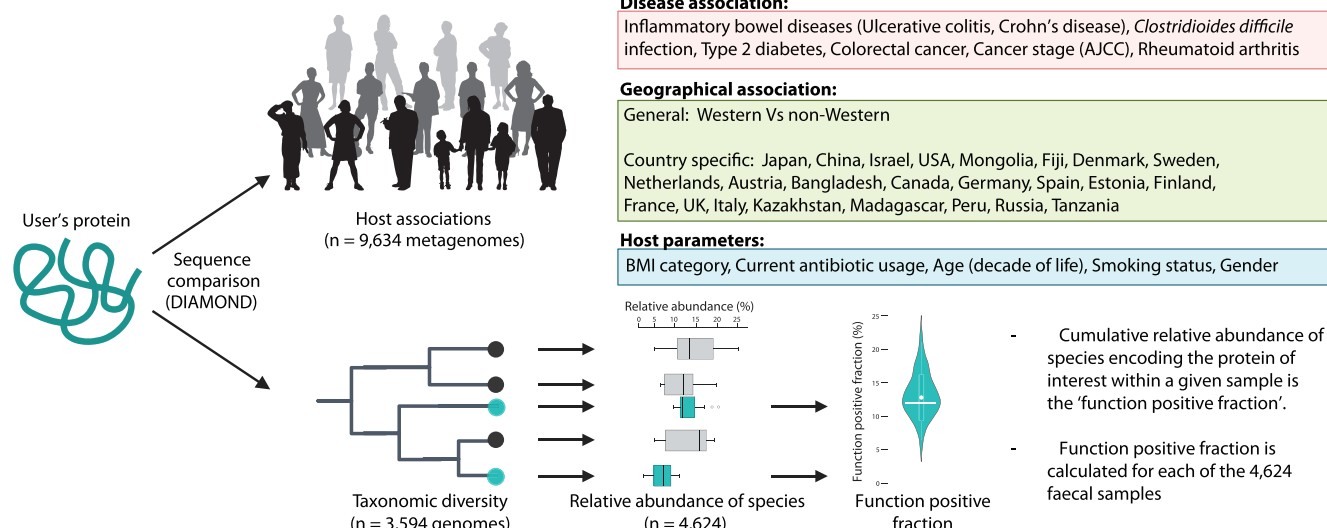

**Disease association:**
Inflammatory bowel diseases (Ulcerative colitis, Crohn's disease), *Clostridioides difficile* infection, Type 2 diabetes, Colorectal cancer, Cancer stage (AJCC), Rheumatoid arthritis

**Geographical association:**
General: Western Vs non-Western

Country specific: Japan, China, Israel, USA, Mongolia, Fiji, Denmark, Sweden, Netherlands, Austria, Bangladesh, Canada, Germany, Spain, Estonia, Finland, France, UK, Italy, Kazakhstan, Madagascar, Peru, Russia, Tanzania

**Host parameters:**
BMI category, Current antibiotic usage, Age (decade of life), Smoking status, Gender

- Cumulative relative abundance of species encoding the protein of interest within a given sample is the 'function positive fraction'.

- Function positive fraction is calculated for each of the 4,624 faecal samples

**Fig. 4 | Workflow of InvestiGUT.** An overview of InvestiGUT, detailing the host associations that are studied, as well as detailing the process for calculation of the function-positive fraction. Functional positive species and their relative abundances are coloured turquoise, along with the final functional positive fraction. Methyl-co-reductase was used as a use-case of the multiple-protein option within InvestiGUT, identifying the complexes prevalence across host age.

sequences are reported. The input sequences are compared against all 846,619,045 predicted protein sequences, representing those from both the metagenomes and genomes. The sequence similarity for annotation is by default 90%, but can be defined by the user. Integration of metadata for the metagenomes is used to quantify the prevalence of the queried proteins with disease and geographic locations, as well as host parameters. The inclusion of genomes from human gut commensals allows the taxonomic range of the protein to be determined. Additionally, we can determine the 'function-positive fraction', which represents the proportion of an individual's microbiota that is positive for the function/protein sequence of interest. This is calculated by determining which genomes encode the protein sequence of interest and then determining the cumulative relative abundance of these genomes across the gut metagenomes of 4464 individuals from the Netherlands and Israel for which pre-computed relative abundance values are available[34].

Methanogenesis in the human gut is restricted to archaea and has been linked to several human diseases[50]. Methyl-coenzyme M reductase (MCR) catalyses the formation of methane in methanogens and consists of an alpha, beta, and gamma chain. Due to MCRs multiple chains, the multiple sequence approach was used in InvestiGUT, ensuring only those samples in which all three chains were detected to be studied. The prevalence of methanogenic functionality increased with host age (Fig. 5a), confirming previous reports of this association[51]. Increased methane production has been reported in individuals with higher BMI, supporting our observation that MCR was more prevalent in individuals with higher BMI[52] (Fig. 5b). Variation with sex has also been previously observed, although age is a confounding variable in this association[51]. The decreased prevalence of MCR in patients noted as having used antibiotics provides the first metagenomic insight into the observation that antibiotic therapy can eliminate methane in the breath of individuals[53]. This may be due to indirect interactions, such as the dependence of methanogenic archaea on bacteria that are themselves susceptible to antibiotics. The MCR-positive fraction revealed a country-specific association, with individuals from the Netherlands having a higher relative abundance of species positive for MCR than individuals from Israel (Fig. 5c).

Country-specific associations can be explained by variation in diet[54]. An example is the enrichment prevalence of seaweed-degrading porphyranases (*Bp1689*), and agarases (*Bp1670*) in the gut of Japanese individuals, compared to North Americans[2]. The original study was limited to 31 individuals (13 Japanese vs 18 North Americans), hence our analysis expands the study of these proteins across 28 countries. We confirmed that both proteins are most prevalent in Japanese samples (80.4% and 85.6%, respectively), followed by China (29.7% and 29.8%, respectively) (Supplementary Fig 2). Interestingly, 45 USA samples were positive for *Bp1689* and 47 for *Bp1670*, perhaps reflecting changes in diet, travel, or immigration over the last decade.

Given that small proteins are highly expressed in the human gut, we investigated the ecological distribution of the 69 most highly expressed SPCs (Fig. 3d). Of these, we observed that 7 were enriched in inflammatory bowel disease (IBD), 9 in ulcerative colitis (UC), and 8 in Crohn's disease (CD). In particular, '55401|ChengpingW_2017_AS130-raw|Bacteria|11|MetaGeneAnnotator' showed a significant change in prevalence with all three conditions across the studies (Fig. 5d). When we explored this association further, we found that the association with UC was only significant in one of the three studies, although the increased prevalence in UC patients was observed in all three studies (Fig. 5e). Analysis of the 69 SPCs expression in patients with UC and CD patients compared to control patients identified that 31 and 34 proteins respectively were enriched in IBD subtypes, including '55401|ChengpingW_2017_AS130raw|Bacteria|11|MetaGeneAnnotator' (Supplementary Table 10, 11). This association may be explained by the predicted antimicrobial activity of the protein, which targets *E. coli*[55], *K. pneumoniae*[56], and *S. aureus*[57], all of which have been studied as causative agents of colitis.

The largest SPC (*n* = 10,700) was also of interest due to its prevalence. It contained a highly conserved core sequence across all sequences, forming a continuous helix (Supplementary Fig 3a). The taxonomic range of this small protein was restricted to the *Bacteroidaceae*, in which it is prolific, with 76.3% (45/59) of *Bacteroides* spp. containing a matching protein sequence (Supplementary Fig 3b). Functional annotation of the protein suggested that it has antimicrobial activity against *S. aureus* (Supplementary Fig 3c). The prevalence of this protein was associated with Westernised individuals, decreased with antibiotic use, and varied depending on host health status (Supplementary Fig 3d-f). These use cases highlight InvestiGUT's ability to facilitate protein ecology studies, confirm associations previously identified in smaller cohorts, and identify associations uncovered by integrating multiple data sources.

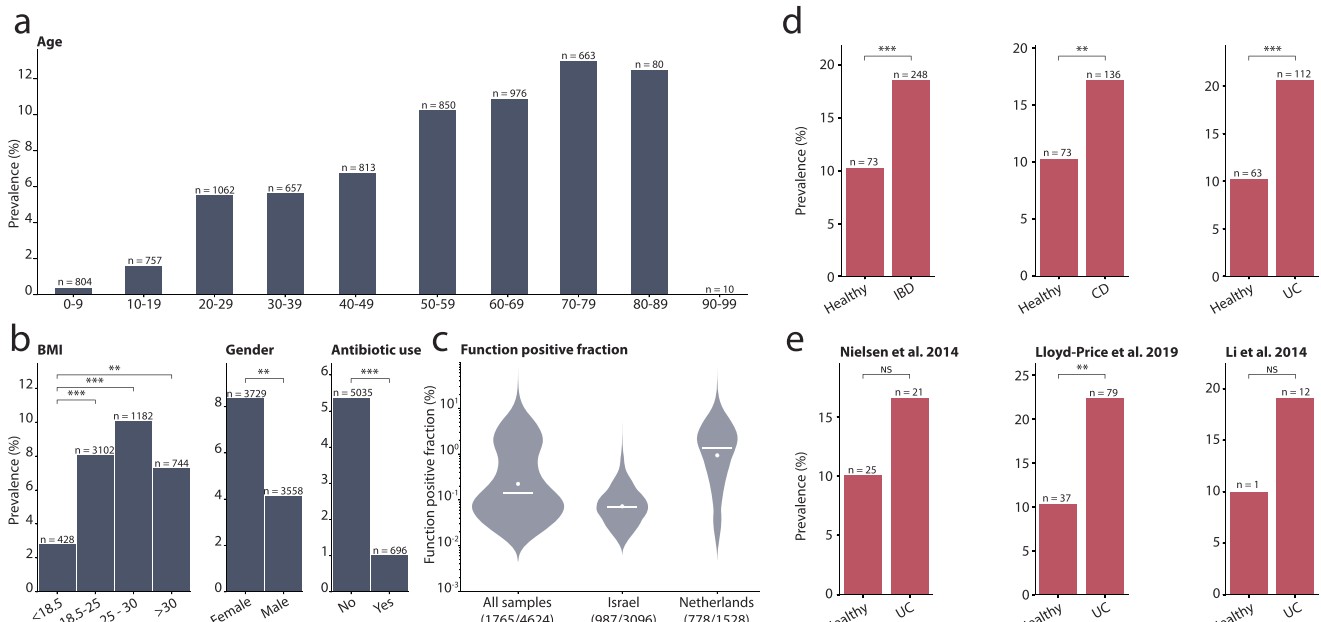

**Fig. 5 | Application of protein ecology to study the association between microbial proteins and host parameters. a** The prevalence of methyl-coenzyme M reductase across metagenomes from patients of different ages. **b** The prevalence of methyl-coenzyme M reductase when grouping samples based on their BMI, sex, and antibiotic use. **c** The methyl-coenzyme M reductase function positive fraction of the microbiome from 4624 individuals as shown by a violin plot which includes a line for the median and a dot for the mean. **d** Significant associations of the small antimicrobial peptide '55401|ChengpingW_2017_AS130raw|Bacteria|11|MetaGeneAnnotator' with IBD, CD, and UC were observed when all studies were combined. **e** To explore the association with UC further, the three individual studies on UC were examined, each showing increased prevalence in UC patients compared to healthy controls. Significance from Fisher's exact test, after Benjamini-Hochberg correction, are shown as: NS not significant, * <0.05, ** <0.01, *** <0.001.

## Discussion

While metagenomes consist of DNA from a range of taxonomic groups, the common practice of using a single gene prediction tool ignores the inherent complexity of these communities. Integrating the taxonomic assignment of a sequence into the choice of gene prediction tools, and optimising the parameters used, allowed us to reveal millions of protein sequences previously missing from human gut gene catalogues. We also observed ~10% variation in the ability of gene prediction tools to correctly predict genes on both fungal and viral genomes. This reinforces the need to select a tool based on the lineage of the sequence being studied. While the variation in perfect predictions between tools was large, the variation was much lower for partial predictions. Interestingly we observed that all tools performed poorly on *Aspergillus nidulans*, with single tools predicting less than 9% of the genes, while a combination of three increased the perfect predictions to 14%. This supports the need for improved tools for eukaryotic gene prediction from metagenomic data[30]. When applied to the human gut, this approach uncovered eukaryotic, archaeal, and viral proteins that have previously been overlooked[27]. While the application of multiple gene prediction tools increased the prediction of correct genes, it also increased the number of spurious predictions compared to catalogues generated with a single tool such as UHGP. As these proteins will not match queried proteins in InvestiGUT or match data aligned to MiProGut this does not pose an issue, but highlights that further improvements are needed to correctly identify genes from microbiomes. To reduce the inclusion of falsely predicted proteins and provide a subset of high-quality protein predictions, we created a filtered version of MiProGut. The first line of evidence studied was the potential for multiple gene prediction tools to predict the same protein at the same location. Therefore, proteins predicted by two tools were retained due to their independent prediction. Secondly, protein clusters containing two or more members were included as they also had independently been predicted, either by the same, or by different

gene prediction tools on separate genomic fragments. Finally, the expression of a protein cluster within one of the 882 metatranscriptomic samples provided the most reliable evidence for the existence of a protein. Based on these, a final collection of 25,266,245 proteins (86.4%) remained within the high-quality protein catalog, MiProGut-HQ. As the selection of gene prediction tools was chosen based on the benchmarking on gut-specific species, further benchmarking is needed before application to additional ecosystems.

Taxonomic assignment of assembled contigs still remains a major barrier. We observed the majority of contigs could not be taxonomically assigned to a lineage to allow for informed gene prediction to occur. It is well documented that further work is required to study the unknown taxonomic diversity within the human gut[20,36,37]. This will enhance the annotated fraction and facilitate a greater understanding of this ecosystem.

The lack of functional annotation by both MANTIS and eggnogmapper for 47.9% of the proteins is consistent with previous observations that nearly half of the identified proteins in the human gut have yet to be characterised[36]. Integration of metatranscriptomic datasets facilitated the identification of highly expressed small proteins with potential antimicrobial activity against key human pathogens. Further analysis of these proteins revealed that the majority (49/69) were differentially expressed between IBD subtypes and non-IBD patients, including those with predicted antimicrobial activity. The association with human health conditions, as well as their antimicrobial activity, warrants further investigation of these proteins to characterise their impact on the microbiota and potential therapeutic application.

The increased coverage of the human gut protein landscape by MiProGut, compared to other gene catalogues, facilitates greater analysis of omics datasets, doubling the aligned fraction in a metatranscriptomic study. While many of the predicted proteins included within MiProGut were singletons, not clustering with any other predicted proteins, transcriptional evidence confirms the existence and

that some are highly expressed, although rarely identified metagenomically. This may be due to them originating from low abundant members of the microbiota, which are often missed by sequencing[58]. As gene catalogues are often used as reference databases for the metagenomic study of an environment, the discovery of these missing proteins will facilitate the identification of disease-specific biomarkers that were previously overlooked. However, the bias of metagenomic samples included towards industrialised countries (USA, China) suggests that the human gut landscape could be further enhanced with the inclusion of additional samples from underrepresented countries and age groups (infants, elderly)[20].

We present the concept of protein ecology, which focuses on the study of proteins rather than taxonomic species and should be applied more broadly to the study of the human gut. InvestiGUT facilitates protein ecology studies by providing statistical analysis of the prevalence of a protein sequence/ sequence of interest across the samples examined in this work. The application of this approach identified both previously observed associations, but across a larger number of samples, providing independent validation of these associations. In addition to protein prevalence, the relative abundance of microbial species containing the queried protein, referred to as the 'function positive fraction', is of interest for protein ecology, as it allows the estimation of protein distribution within the ecosystem. The current genome collection and approach does not account for strain-level diversity, instead assuming that the presence within the reference genome is representative. Currently, this limits our ability to truly capture gut microbiota diversity. Future integration of a larger collection of genomes that captures the strain-level diversity of each species would facilitate the calculation of a modified functional positive fraction.

## Methods

### Selection of optimal gene prediction tools

To test the performance of gene prediction tools for large-scale metagenomic predictions, 13 different tools (Supplementary Table 1) were evaluated individually as well as in combinations of two or three using the ORForise framework[29]. Among these, three tools were designed for eukaryotic sequences[38,59,60], nine focused on prokaryotes[61–70], and one for viruses[71]. Both Prodigal[26] and Pyrodigal (v2.1.0) were included in the comparisons, but as both provided identical results, Prodigal was removed from the comparisons. Three of the eukaryotic tools (AUGUSTUS[38] v3.3, GlimmerHMM[59] v3.0.4, and SNAP[60] v2006-07-28) rely on the selection of a model before genes can be predicted. AUGUSTUS was run with the built-in saccharomyces_cerevisiae_S288C model on all eukaryotic genomes. Both other tools had no default fungal model, so a model was built based on *Saccharomyces cerevisiae* S288C (GCF_000146045.2).

The genomes and corresponding gene predictions used in the tool comparison are detailed in Supplementary Table 2. The bacteria genomes studied included the six model organisms used in the publication  of the ORForise framework[29]. Despite this set consisting only of species from the two phyla Pseudomonadota and Bacillota, the study revealed significant differences in gene characteristics, both unique to individual genomes and across the group, that are critical for accurate gene prediction. These included substantial variations in start codon usage, gene length and overlap, GC content, and the utilisation of the 'canonical' stop codon TGA, which in some genomes and even niche genes, serves as a codon for tryptophan. These characteristics posed considerable challenges to the 15 gene prediction tools compared in the study, with no single tool able to fully overcome them. To address these limitations, we expanded the dataset by including additional bacterial species known to be either pathogenic[72–76] or health-associated[77–82]. The final set included bacterial ($n = 17$), archaeal ($n = 3$), eukaryotes/fungal ($n = 3$), and viral ($n = 3$) genomes (total of 26 species), all selected from strains associated with the mammalian gut.

For the fungi, only nuclear genomes were considered to prevent the inclusion of non-specific mitochondrial sequences.

The under- and overprediction of the tools was quantified using ORForise[29], which provides metrics on the performance of gene prediction software, by comparing predicted results to reference annotations (Supplementary Table 3).

### Datasets

A total of 9677 metagenomes from 43 studies were analysed (Supplementary Table 4). Among those, 7735 metagenomic assemblies from 41 studies were available from Pasolli et al.[8] and 1326 assemblies from the Integrative Human Microbiome Project[9]. FASTQ files for 616 metagenomes from a Japanese cohort[83] were also downloaded and processed. Human sequences were removed from the latter metagenomic reads by running BBMap[84] (v38.18) with Genome Reference Consortium Human Build 38 (GRCh38) as a reference. Reads were assembled using MEGAHIT[85] (v1.2.9) with a range of k-mer sizes from 21 to 99, and a minimum count of 5 to filter out low-quality reads. Metagenomes that encode less than 1000 proteins predicted were removed ($n = 43$) due to a lack of sequencing information, meaning 9634 metagenomes were retained. Metadata for metagenomic samples was either obtained directly, or from the curatedMetagenomicData R package[86]. Additionally, a collection of 3594 non-redundant high-quality genomes from Leviatan et al.[34] were studied, this included both metagenomic assembled genomes (MAGs) and isolate genomes. These non-redundant genomes had a reported median completeness of 95% and median contamination of 0.67%[34].

### Taxonomic assignment

For the taxonomic classification of all contigs, Kraken 2 (v2.1.2)[87] was applied with a confidence threshold of 0.15[88]. To enhance the taxonomy-assigned fraction of each metagenome, a custom Kraken 2 database was formed using all complete genomes for Archaea, Bacteria, Eukaryota, and Viruses from the NCBI RefSeq database (April 2023)[32]. The inclusion of the human genome (Genome Reference Consortium Human Build 38 (GRCh38)) within the Kraken 2 database facilitated the removal of host contamination from downstream gene prediction. Based on the taxonomic ID assigned to each contig, the correct genetic code was assigned according to the NCBI Taxonomy Database (August 2023). For representative genomes, the taxonomic classifications of Leviatan et al. were computed with GTDB-Tk (v2.3.2; r214)[34,89]. Contigs from each metagenome/genome were subdivided into files based on their inferred taxonomy and genetic code with the selected combination of three gene prediction tools being run on the corresponding subset files. Contigs without an assigned taxonomy were annotated with the bacteria-optimised selection of tools and genetic code 11.

### Gene prediction

Variability in tool outputs was accounted for with a script to parse the start, stop, and strand information of prediction outputs. When multiple tools predicted genes with the same stop position, but different start positions, the longest sequence was selected. In cases of differing stop codons, all sequences were retained. Secondly, all partial genes were removed from consideration, including those with a start codon in the first or last three bases of a contig, where assembly errors are common[90], to prevent the inclusion of truncated genes. When possible, the minimum prediction length setting was set to 21 bases (six amino acids plus a stop codon).

Based on this information, protein sequences were generated using the genetic code corresponding to the Kraken 2 taxonomy ID predictions. No protein count threshold was applied for the number of predicted genes in the studied representative genomes.

Protein sequences are named using a standardised system of metadata connected by the '|' symbol. The names start with a unique

number for each protein predicted from a single file. Next, the name of the dataset the predicted protein is from is included. The domain the originating contig was assigned to is then stated, followed by the codon table which was used. Finally, the name of the gene prediction tools that predicted the protein are provided, if multiple tools predicted the same protein, then all tools are stated connected by hyphens. An example is "55401|ChengpingW_2017_AS130raw|Bacteria|11|MetaGeneAnnotator" which is the 55,401st protein predicted from the dataset "ChengpingW_2017_AS130raw". The contig it was predicted from was assigned as Bacterial, and codon table 11 was used by MetaGeneAnnotator for its prediction.

### Protein ecology across human gut samples (InvestiGUT)

InvestiGUT (v1.1) accepts either a single protein sequence or a set of sequences, as defined by the parameters '−s' or '−m' respectively. User-provided protein sequences are matched to the collection of human gut microbiome proteins via DIAMOND (v2.1.11)[91] alignment with a default minimum query and subject coverage of 90% and identity of 90%, which can be defined by the user. User proteins can be analysed individually (-s), or as a group (-m) where all proteins must be present within a sample to be considered. The group analysis allows users to determine how frequently all enzymes in a certain pathway or all subunits in a protein complex are present. Output is divided into metagenomic and species-based analysis results, with both being available as raw data in the form of 'tab separated value' files and as automatically generated vector figures.

Metadata for each metagenomic sample included age, sex, BMI, smoker-status, westernised-diet status, use of antibiotics, and geographical location. Additionally, disease prevalence includes output for CD, ulcerative colitis, colorectal cancer, *Clostridioides difficile* infection, type-2 diabetes, rheumatoid arthritis, and fatty liver disease. Metadata vocabulary is consistent with that used in the curatedMetagenomeData repository. Prevalence of the queried sequence in each of these groups is quantified, and compared either between categories, or healthy controls, using Fisher's exact test from the scipy.stats[92] module in Python.

Querying the user input protein against the 3594 species representative genomes provides a list of species which contain the protein, termed "function-positive species". The taxonomic range of the protein is determined by the prevalence of the protein with each genus, family, and phylum of positive species. The function-positive fraction is obtained as the cumulative relative abundance of function-positive genomes across 4624 individuals for which precomputed relative abundance values are available. InvestiGUT is available at: https://github.com/Matt-Schmitz/InvestiGUT.

### Gene catalogue creation

The protein sequences from the 9634 filtered human gut metagenomes and 3594 genomes were clustered using MMseqs2 (v14-7e284)[93] linclust using parameters '--cov-mode 1 -c 0.8' (minimum coverage threshold of 80% the length of the shortest sequence), '--k-mer-per-seq 80' (number of k-mers selected per sequence), and '--min-seq-id 0.9' to cluster at 90% protein identity. Thresholds were selected as previously used to generate the UHGP-90 to allow for comparison, with additional clustering conducted at 50%, 95%, and 100% protein identity. Overlap between the MiProGut-90 and UHGP-90 was identified by combining the proteins of both catalogues and reclustering with the same parameters.

### Functional annotation

The MiProGut-90 gene catalogue was annotated using eggNOG-mapper (v2.1.12) with the inbuilt DIAMOND blastp search option and default parameters[94,95]. Further annotation was conducted with MANTIS[96] using the integrated annotation file and default parameters. When proteins had more than one functional category or general category letter association, the corresponding functional descriptions of each category were evenly weighted. Antimicrobial activity was predicted using the DBAASP toolkit[49]. Positive activity against *Escherichia coli* ATCC 25922, *Pseudomonas aeruginosa* ATCC 27853, *Klebsiella pneumoniae*, *Staphylococcus aureus* ATCC 25923, *Candida albicans*, and *Saccharomyces cerevisiae* was determined as a predicted MIC < 25 µg/ml. Annotation against the NCBI nr database was conducted using BLASTP with default parameters[97].

### Metatranscriptome expression analysis

DIAMOND (v2.1.11)[91] mapped metatranscriptomic reads[7,42,43] against the MiProGut-90 and UHGP-90 protein catalogues using the 'blastx' command with parameters '--id 90 --evalue 1e-6 -k 1 --max-hsps 1'. The aligned fraction was calculated per sample as: (aligned reads/total reads) × 100. Expression of specific proteins was quantified as reads per kilobase per million mapped reads (RPKM) and calculated using: (aligned reads/(gene length/1000) × (total reads/1,000,000)).

### Statistics and reproducibility

No statistical method was used to predetermine sample size as data was gathered from multiple studies, and only those datapoints with insufficient data (<1000 proteins) were excluded from the analyses.

### Reporting summary

Further information on research design is available in the Nature Portfolio Reporting Summary linked to this article.

## Data availability

The datasets used in this study are described in Supplementary Table 4. Gene and protein predictions, along with the non-redundant MiProGut catalogue at 50%, 90% (termed MiProGut throughout the paper), 95%, and 100% protein identity, as well as the high-quality filtered version (MiProGut-HQ) are available at: https://doi.org/10.5281/zenodo.10988030.

## Code availability

InvestiGUT, along with the protein prediction pipeline used in this work, is available at: https://github.com/Matt-Schmitz/InvestiGUT (https://doi.org/10.5281/zenodo.14973403).

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

## Acknowledgements

The authors would like to thank Susan Jennings and Charlie Pauvert for reviewing the paper prior to submission, and Amy Coates for designing the logo. TCAH received funding from the RWTH Start-up grant, titled ProtoBIOME, and the University Hospital of RWTH Aachen START grant, titled LeakyGut (021/23). TC received funding from the German Research Foundation (DFG): project no. 403224013—SFB1382 Gut-liver axis, subproject Q02; project no. 460129525—NFDI4Microbiota. The silhouettes of people come from https://all-free-download.com/.

## Author contributions

T.C.A.H. and M.A.S. designed the project. T.C.A.H. supervised the project, with data analysed by M.A.S. N.J.D. contributed to discussions on the design of experiments, optimisation, and provided expertise on benchmarking of gene prediction tools. T.C. provided infrastructural support and all authors wrote and reviewed the manuscript.

## Funding

## Competing interests

The authors declare no competing interests.
