## [Transparent Peer Review file · Nature Communications]

Lineage-specific microbial protein prediction enables large-scale exploration of protein ecology within the human gut

Corresponding Author: Dr Thomas Hitch

Version 0:

Reviewer comments:

Reviewer #1

(Remarks to the Author)

The authors of the manuscript „Lineage-specific microbial protein prediction enables large-scale exploration of protein ecology within the human gut” address two important problems concerning microbial proteins. (1) They create MiProtGut by cataloging proteins produced by human gut microbiota of all domains of life. (2) They propose and demonstrate the utility of InvestiGUT – a computational tool to associate protein prevalence with metadata in meta-omics studies. Although such catalogs are currently available (e.g. UHGP) they cover only bacterial proteins, which significantly hampers our functional understanding of microbial communities. The authors propose an optimized procedure of gene prediction based on a taxonomic assignment of genomes of interest as well as the method of finding proteins of interest in the newly created gene catalog. Overall, it is an impressive and expansive work with a high relevance to human microbiome research.

Major comments

1. In the Taxonomic assignment part, the contains are assigned to a different taxonomy (NCBI) than genomes from Leviatan et al. (GTDB). Is there a particular reason why two different taxonomies were used for different data?
2. In the context of taxonomy assignment, the authors also gloss over the significant “Unknown” category (close to 50% of the entire MiProtGut collection), which may be a consequence of the choice of method (Kraken) and the database (NCBI). Including a wider discussion of this issue or an attempt to solve it entirely would be welcome.
3. In creating the MiProtGut (redundant) collection the authors seem to concatenate all predictions from all methods, regardless of the level of agreement between different gene predictors (adding the numbers in Fig. 1D), then cluster everything at 90% sequence identity. This process is not very clearly explained in the text (or even contradicted in lines 236-239), it also seems to defy the purpose of the whole consensus approach and increase the likelihood of false positives. I would welcome a deeper discussion of these issues in the text, in particular with relation to the possible false positives, metatranscriptomic analyses, comparisons to UHGP and identification of short peptides. Also, MiProtGut is likely to be a widely used resource, I would welcome making more versions of it available for download (e.g. raw, clustered at 100%; similarly to UHGP).
4. The title of the paper states „Lineage-specific microbial protein prediction...” which is certainly true, however I am missing a more in-depth discussion of how/if this lineage-specific approach improved the results versus how other choices (using multiple gene prediction tools, considering short peptides, filtering and merging) impacted the MiProtGut collection.
5. The authors put an effort into ensuring that newly identified proteins are not artifacts of the gene prediction methods. One of the ways to do so was to analyze the overlap with UHGP. Another, was a metatranscriptomic validation showing a higher coverage of metatranscriptomics data on 3 datasets compared to UHGP. However, this raises several issues downstream.
 - 5.1. UHGP consists solely of prokaryotic proteins, therefore its use to validate the prediction of eukaryotic proteins is questionable. In such a way it might be useful to study the overlap between datasets with respect to different taxonomic groups eg. Bacteria, Archaea, etc.
 - 5.2. UHGP is built using a partially-overlapping set of source datasets which may explain why 3.3M proteins don't overlap with MiProtGut. However, the authors never discuss this significant set of proteins (~25% of UHGP-90) – can this be explained by different source data alone?
 - 5.3. the MiProtGut collection contains a significant number of singletons (14M; line 300). It is not shown/discussed if

singletons are shared between UHGP and MiProtGut or if they are unique. In the latter case, this increases the risk of the dataset being populated with false positives.

6. I highly appreciate the use of metatranscriptomic data to further validate the predicted proteins. In this case, it would also be interesting to look at the taxonomic origin of proteins aligned to metatranscriptomic data to see if the increase in coverage is due to newly identified eukaryotic or prokaryotic proteins.

7. Small proteins are indeed a very overlooked group of proteins however, it is not without a reason. Many gene prediction tools reject short sequences to keep the false positive ratio low. Here authors modified the parameters of gene predictors to include such proteins. Unfortunately, this part is not so well described in the methods section. Furthermore, such a change might affect the false positive ratio of such methods. It would be beneficial to check how such change will affect the analysis or, at least, discuss it in the main text.

8. The InvestiGUT is certainly a valuable tool, nevertheless it would benefit from extending the Readme section within the GitHub repository. Especially adding a short paragraph at the beginning describing the purpose of the tool could be helpful to users who did not read the text carefully.

9. In Figures 3 and 4 the fonts are tiny and difficult to read. I am open to the authors making the figures more legible in whichever way they see fit. My suggestion for Figure 3 would be to parse names in panel D to make them shorter and easier to read. For Figure 4, I would suggest splitting it into 2 figures – panel A as a figure, then panels B-F a separate figure. Some more minor comments about Figs. 3 and 4:

- Fig. 3A: top bar shows length 0-50. In fact, according to the Methods section the minimum length is 6 residues
- Fig. 3C: there are no units given for the Y-axis (should it be %?)
- Fig. 4E: There are no studies identified as data sources. Are those the same studies as in panel (f)?
- Fig. 4 caption: The caption is incompatible with previous captions. For example, it moves defining the panel (a), (b), etc. to ends of sentences. Also, it describes some results in full sentences, which is fine, but also inconsistent with previous captions.

Minor comments

- abstract: The MiProtGut collection is not mentioned in the abstract at all – I would welcome including this important mention in the abstract
- l. 68: has -> was
- l. 70: other taxa -> other domains (or clades?)
- l. 87-90: explanation of why Pyrodigal was used is phrased in a confusing way. It would be sufficient to say that Pyrodigal and Prodigal give identical results (l. 88-89), hence only one was used.
- l. 99-101: the authors evaluate 26 species from different domains of life but no Eukaryotes. Why? Is there a specific reason for that?
- l. 128-130: in lines 111-113 the authors mention removing the human genome data using BMap, then in lines 128-130 they do it again, but using Kraken2. Was that necessary? If yes, I welcome a comment on why it was necessary and some data on how much improvement did the second pass-through using Kraken2 provided
- l. 148: six amino acids plus stop codon -> six amino acids plus a stop codon
- l. 154: Convention suggests that upon release v. 1.0 is used. Why InvestiGUT is still in v. 0.1?
- l. 154-164: This paragraph includes a very technical description of the InvestiGUT package. Most of it could be moved to the README section of the repository without detriment to the publication.
- l. 192-201: the paragraph discusses functional annotations but there is no mention of the ontology that is being produced. Is GO or KEGG used? Or are those just headers from a protein database (which?)
- Figure 1A: dereplicated suggests removing redundancy (clustering at 100% sequence identity), while the number of genes termed „dereplicated” corresponds to the MiProtGut-90 (clustering at 90% seq. id.)
- l. 207-210: equations in their current form are hard to read, especially the equation in line 210
- Figure 2B: The scale on the Y-axis is missing. As is, it reads that the scale goes up to 3 protein clusters. Also, there is a break between the end of the grey color bar and 3.0 – what is the actual maximum?
- l. 275: refraction -> rarefaction
- l. 276: functional annotation shown here is from eggNOG-mapper, MANTIS, DBAASP, or a sum of all?
- l. 292: permeations -> permutations (?)
- l. 319: it's unclear to me why the number of clusters is expressed as integers and not natural numbers
- l. 322-323: should this be the final sentence of the previous paragraph? Also, there is a connecting sentence missing linking the analysis of MiProtGut-UHGP overlap to the analysis of metatranscriptomic data
- l. 324: Figure 2c -> Figure 2d
- l. 392-393: phrase „where only samples that are positive for all queried sequences are reported” was unclear to me until I reached the subsequent paragraph. Please, consider rephrasing.
- l. 404: who are the 4,464 individuals? The phrasing is unclear whether this is only the case in the particular analysis the authors describe, or if it is a feature of InvestiGUT that it always performs a comparison against an arbitrary set of individuals.
- l. 415: gender -> sex
- l. 437: are the 69 SPC the same as in Figure 3C? Maybe reference?
- l. 440 & 447: encoding of protein identifiers is never explained in the text and hard to read in its current format

(Remarks on code availability)

I checked the repository but didn't run the code. Some comments about the code are included in the comments to the

authors.

Reviewer #2

(Remarks to the Author)

(Remarks on code availability)

Reviewer #3

(Remarks to the Author)

The manuscript "Lineage-specific microbial protein prediction enables large-scale exploration of protein ecology within the human gut" by Schmitz et al. describes an important issue in the field of metagenomics, the problem that most metagenome workflows rely on general purpose gene prediction instead of incorporating a lineage-specific ORF prediction on assembled contigs. They present a new method, InvestiGUT, that enables the use of the correct genetic code based on taxonomic assignment of genomic fragments, removes partial predictions and incorporates the prediction of small proteins. The tool itself is easy to install and runs the given example correctly after following the installation instructions.

We feel that the manuscript has some major issues that need further revision:

- The introduction and discussion lack focus on the proposed contribution: Why are we struggling to predict correct ORFs? What could be done instead? Why this is an overall problem? Can we quantify/estimate the problem of incorrectly predicted genes in public databases or MAG/gene catalogs? What are the current approaches to overcome this or why lineage-specific prediction is a necessary approach that needs to be implemented
- In order to prove its usefulness for metagenomic applications, benchmarking of the ensemble gene calling approach should include more organisms (if well-annotated representatives are available) covering a wide variety of genetic codes and/or species with peculiarities, e.g. alternative start codons (e.g. Actinobacteria).
- Most importantly, the ensemble ORF prediction workflow, which is currently not part of the git repository should be available as tool/pipeline:
 - o Where is the prediction workflow code?

Minor points:

- There should be functionality to build your own catalogue.
- The reason for the sequence identity thresholds chosen for clustering during catalogue creation should have a function/mechanism-driven explanation, rather than just being the ones used in UHGP-90.
- Parameters for eggNOG-mapper/MANTIS are missing (or please mention if all are default).
- In the Supplement for the prediction benchmark, please include a column with the percentage of spurious predictions?
- Regarding the ORF prediction benchmark results from ORForizer and merging 1, 2, and 3 tool combinations:
 - o For all tools, relying on a single tool with the 'best' results (lowest spurious rate) seems to lead to much more favorable real ORF to spurious ratios.
 - o Even when using three tools, the 'best' combination in terms of minimizing spurious predictions was not chosen for catalog generation. Instead, the combination that gave the most predictions was selected, even though the benchmark data showed an extreme inflation of spurious predictions compared to the tool with the lowest spurious ratio. This observation is based on the modified and merged results tables per domain in '515720_0_supp_652116_sdxq81_review_mod.ods'. It is critical to clarify the reasons for this choice, as it could have a significant impact on the quality and reliability of the resulting catalog.
- Inconsistency of tools used across domains in Figure 1: There appears to be a mismatch between the tools used for each domain in section (a) and the tools compared per domain in section (d). Specifically, it seems that the tool sets for viruses and archaea may have been mixed up.
- The overlap with UHGP-90 doesn't prove non-spuriousness of predictions since it was created using prodigal and cannot be claimed to be error free either.
- The use of metatranscriptomics data as a control is a good approach. It could be used to filter out predicted ORFs with no metaT support, which are then likely to be spurious (and perhaps could also be used to calibrate the ensemble ORF prediction approach on real-world data).
- The small protein and AMP parts of the manuscript are a nice side tangent of the project but feel slightly disconnected with the 'main' part, i.e. the prediction workflow and its validation (and subsequently the validity of the catalog).
- Some suggestions to make the tool more user-friendly and runnable on large-scale datasets
 - o Allow users to control the number of threads for DIAMOND search
 - o Block size to optimize search time
 - o E-value for catalog search is hardcoded to 1e-3; make it user-adjustable?

(Remarks on code availability)

See main review, the installation and the example is working, the code for the prediction is missing.

Reviewer #4

(Remarks to the Author)

(Remarks on code availability)

Version 1:

Reviewer comments:

Reviewer #1

(Remarks to the Author)

I want to express my gratitude to the authors for their meticulous attention to the issues raised in my feedback. All of the minor comments have been satisfactorily addressed. Major comments, in principle, are also well-addressed.

The revision process has also helped me gain a deeper understanding of the Authors' intentions behind this work. Considering that the primary research outcome of this work is the InvestiGUT, I would like to suggest that the Authors consider how the MiProGut collection is presented and discussed.

In particular, MiProGut is likely to be perceived as an alternative to the UHGP collection and be used also outside the InvestiGUT framework. While UHGP may be incomplete, some effort by the authors of UHGP work was put into minimizing the number of false positives. Since MiProGut relies on a sum of different predictions, it is possible that it may have more false positive genes, even though the Authors provide some reassurance that this may not be a significant issue through the metatranscriptomics case-study.

To address this, I propose some alternative solutions:

1. Provide a broader discussion about the nature and limitations of MiProGut in the paper (ideally also by including appropriate notice in READMEs). For instance, you could expand upon the existing discussion in lines 562-581.
2. Create a MiProGut-HQ (high-quality) collection by intersecting gene predictors or employing a different "false positive" detection method. This approach would make the collection more appealing to researchers studying microbial proteins in other non-omics contexts where the issue of false positives could be particularly significant.

(Remarks on code availability)

I did code assessment during my first round of reviews.

I did not run the code this time around, but I did study the changes made by the Authors since the last review.

Reviewer #3

(Remarks to the Author)

The authors have responded sufficiently to all our comments. Only problems with the git, see below Thanks

(Remarks on code availability)

Tested on Macbook Apple M2 Pro, 16 GB, Sequoia 15.2, with the given command

```
mamba create --no-channel-priority -n investigut > -c bioconda -c conda-forge > "python=3.11" "numpy=1.24.3"
"scipy=1.10.1" > "conda-forge::matplotlib-base" "seaborn=0.13.0" > "pandas=1.5.3" "statsmodels=0.13.5" "ete3=3.1.2" >
"openpyxl=3.0.10" "bioconda::diamond=2.1.8"
```

I get the following error:

```
error libmamba Could not solve for environment specs
```

```
The following package could not be installed
```

```
└─ diamond =2.1.8 * does not exist (perhaps a typo or a missing channel).
```

Reviewer #4

(Remarks to the Author)

(Remarks on code availability)

Response to reviewers

The authors of the manuscript „Lineage-specific microbial protein prediction enables large-scale exploration of protein ecology within the human gut” address two important problems concerning microbial proteins. (1) They create MiProtGut by cataloging proteins produced by human gut microbiota of all domains of life. (2) They propose and demonstrate the utility of InvestiGUT – a computational tool to associate protein prevalence with metadata in meta-omics studies.

Although such catalogs are currently available (e.g. UHGP) they cover only bacterial proteins, which significantly hampers our functional understanding of microbial communities. The authors propose an optimized procedure of gene prediction based on a taxonomic assignment of genomes of interest as well as the method of finding proteins of interest in the newly created gene catalog. Overall, it is an impressive and expansive work with a high relevance to human microbiome research.

Response: We thank the reviewers for seeing the value in our work and for taking the time to review our manuscript. Below we detail how we have addressed each of the reviewers comments, which we believe has improved the manuscript substantially.

Major comments

1. In the Taxonomic assignment part, the contigs are assigned to a different taxonomy (NCBI) than genomes from Leviatan et al. (GTDB). Is there a particular reason why two different taxonomies were used for different data?

Response: We understand the reviewers reservation about using two different taxonomic databases within this work. The NCBI taxonomy was only used for initial determination of each contigs genetic code, as the NCBI provide the resource that links genetic codes to taxon IDs, we used their taxonomic assignments. However, GTDB provides a more accurate look at taxonomy when studying prokaryotes, hence for the prokaryote genomes, we chose to use GTDB. GTDB also provides placeholder names for many taxa not recognised within the NCBI taxonomic system.

2. In the context of taxonomy assignment, the authors also gloss over the significant “Unknown” category (close to 50% of the entire MiProtGut collection), which may be a consequence of the choice of method (Kraken) and the database (NCBI). Including a wider discussion of this issue or an attempt to solve it entirely would be welcome.

Response: We agree with the reviewers that we glossed over this aspect, this is because we are normalised to this range of unknowns. As shown by Thomas and Segata (<https://doi.org/10.1186/s12915-019-0667-z>) a large fraction of metagenomic data from the gut lacks a match to any genome, and many of those that do match are to MAGs, which are themselves inferred. Due to this we made the choice to filter the NCBI database to only include complete genomes, reducing the chance for false positive assignments. To expand the discussion on this topic sufficiently we annotated all the contigs with available ‘default’ database (core_nt) to compare the unannotated fraction. These results can be seen in **Supplementary Table 6**. We also discuss the lack of improvement in the discussion. In brief, no significant improvement was observed, hence the ‘complete genome filtered’ custom database was retained during

analysis. We also chose a confidence value of 0.15 based on previous publications and discussion on the Kraken GitHub that the common use of 0 is insufficient to prevent false positive matches. This leads to a higher percentage of unassigned contigs, but ensure confidence in those that are assigned.

3. In creating the MiProtGut (redundant) collection the authors seem to concatenate all predictions from all methods, regardless of the level of agreement between different gene predictors (adding the numbers in Fig. 1D), then cluster everything at 90% sequence identity. This process is not very clearly explained in the text (or even contradicted in lines 236-239), it also seems to defy the purpose of the whole consensus approach and increase the likelihood of false positives. I would welcome a deeper discussion of these issues in the text, in particular with relation to the possible false positives, metatranscriptomic analyses, comparisons to UHGP and identification of short peptides. Also, MiProtGut is likely to be a widely used resource, I would welcome making more versions of it available for download (e.g. raw, clustered at 100%; similarly to UHGP).

Response: We believe there is a misunderstanding of the approach we apply so have attempted to rectify this in the text. Firstly, as stated by the reviewer, we never use a consensus approach to merge the annotations from the three tools. Our goal was to ensure all predicted proteins are captured by our approach. With this in mind we wanted to reduce the chance of spurious predictions, but given the variability in proteins predicted between tools, we chose to utilise multiple tools and then filter low quality matches i.e. incomplete genes or shorter predictions of the same protein, rather than limit ourselves to a single tool. To clarify this we have removed the use of the term 'consensus' from the legend of Figure 1. We have also included additional text regarding the limitation of this approach:

“While this approach does result in an inflation of spurious predictions, below we have applied multiple approaches to study the benefit of this approach. This includes metatranscriptomic analysis to identify evidence for expression, and comparison with an independent gene catalogue to provide independent validation. Based on these results, we believe it is more advantageous to predict additional real proteins at the expense of including spurious genes, than to risk exclude real proteins.”

These sections are now all inline with the lines mentioned by the reviewer, and we hope it is clearer that we predict genes with all three methods and accept each as a valid gene prediction.

Regarding the point of false positives, this is true, our approach most definitely leads to a greater number of spurious protein predictions. However, in InvestiGUT these proteins would never be matched, so do not impact the analysis, and within the protein catalogue they would not be assigned a function or align to metatranscriptomic data, so would be inconsequential. Therefore we believe this approach has far more benefits than downsides. We have expanded the discussion to reflect this as well:

“While the application of multiple gene prediction tools increased the prediction of correct genes, it also increased the number of spurious predictions. As these proteins will not match queried proteins in InvestiGUT or match data aligned to MiProGut this

does not pose an issue, but highlights that further improvements are needed to correctly identify genes from microbiomes. As the selection of gene prediction tools was selected based on the benchmarking on gut specific species, further benchmarking is needed before application to additional ecosystems.”

Regarding the alternative versions of MiProGut, we have now generated a raw version, a 50%, 90%, 95%, and 100% versions to maximise their use. These are linked in the paper.

4. The title of the paper states „Lineage-specific microbial protein prediction...” which is certainly true, however I am missing a more in-depth discussion of how/if this lineage-specific approach improved the results versus how other choices (using multiple gene prediction tools, considering short peptides, filtering and merging) impacted the MiProtGut collection.

Response: As the reviewer points out, we have tweaked and improved gene prediction by multiple methods within this paper, and lineage-specific annotation is but one approach. To expand upon the impact of selecting tools based on the taxonomic lineage of a metagenomic sequence we have included the following section to the discussion:

*“While metagenomes consist of DNA from a range of taxonomic groups, the common practice of using a single gene prediction tool ignores the inherent complexity of these communities. Integrating the taxonomic assignment of a sequence into the choice of gene prediction tools, and optimising the parameters used, allowed us to reveal millions of protein sequences previously missing from human gut gene catalogues. We also observed ~10% variation in the ability of gene prediction tools to correctly predict genes on both fungal and viral genomes. This reinforces the need to select a tool based on the lineage of the sequence being studied. While the variation in perfect predictions between tools was large, the variation was much lower for partial predictions. Interestingly we observed that all tools performed poorly on *Aspergillus nidulans*, with single tools predicting less than 9% of the genes, while a combination of three increased the perfect predictions to 14%. When applied to the human gut, this approach uncovered eukaryotic, archaeal and viral proteins that have previously been overlooked²⁷.”*

Due to the nature of metagenomes not being derived from a known set of microbes we cannot determine the effect on metagenome predictions but believe the genome based results provide interesting insight.

5. The authors put an effort into ensuring that newly identified proteins are not artifacts of the gene prediction methods. One of the ways to do so was to analyze the overlap with UHGP. Another, was a metatranscriptomic validation showing a higher coverage of metatranscriptomics data on 3 datasets compared to UHGP. However, this raises several issues downstream.

5.1. UHGP consists solely of prokaryotic proteins, therefore its use to validate the prediction of eukaryotic proteins is questionable. In such a way it might be

useful to study the overlap between datasets with respect to different taxonomic groups eg. Bacteria, Archaea, etc.

Response: We agree with the reviewer that while the UHGP is the most up to date gene prediction catalogue, it is specific to prokaryotes, highlighting again the need to consider eukaryotes and viruses. To better compare against the UHGP, we have studied the protein clusters represented by each domain subset of sequences from MiProGut, and redone the analysis. The results from this are included in **Supplementary Table 8**. This analysis has been very interesting as it highlights the variation in original sources between the two catalogues as the Bacterial, Archaeal, and unknown proteins matched at ~40%, but only 16% of viral proteins and 0.3% of eukaryotic proteins matched the UHGP. This is likely due to UHGP being bacterial specific, although likely also included Archaeal MAGs. This has been included within the results:

*“The UHGP proteins not captured by MiProGut are likely due to the different sources each catalogue used to predict proteins, namely the focus on genomes for UHGP and metagenomes for MiProGut. As UHGP was generated from prokaryotic genomes, the majority of MiProGut proteins matching UHGP were of bacterial, archaeal, or unknown origin. Conversely, only 16.6% of viral and 0.3% of eukaryotic MiProGut proteins matched UHGP due to their exclusion (**Supplementary Table 8**).”*

We also identified many protein clusters which contained a mixture of origins. The existence of clusters containing members from different lineages opens the possibility for horizontal gene transfer to be studied. Due to this, we expanded the analysis and have written an addition section and supplementary Figure on this topic:

*“The inclusion of genomes representative of all prokaryotes detected in the human gut allowed us to investigate the occurrence of protein clusters across different phylogenomic distances. Of the 29,232,514 protein clusters in MiProGut, 19.6% (5,734,391) were detected within a genome, and only 927,384 (3.2%) occurred in genomes assigned to multiple species (**Supplementary Figure 1a**). To better assess the taxonomic range at which each protein cluster is shared, we evaluated the genus (**Supplementary Figure 1b**), and phylum level (**Supplementary Figure 1c**) and observed 142,342 and 19,320 protein clusters to be shared across multiple taxa at each level, respectively. To identify the most frequently shared protein clusters, we filtered for those present in ≥ 10 species and ≥ 2 phyla. Of these 2,423 clusters, 2,197 (90.6%) were present in Firmicutes_A, a division of the phyla Bacillota (formerly Firmicutes) (**Supplementary Figure 1d**). As viruses are a known mediator of horizontal gene transfer within the gut ⁸³ we investigated the protein clusters containing a protein of viral origin ($n = 77,496$) that also occurred in the reference genomes ($n = 2,914$) (**Supplementary Figure 1e**). Of these, 75 protein clusters occurred in at least two phyla and were annotated as including virulence and antibiotic resistance genes (**Supplementary Figure 1f**). The most promiscuous viral protein cluster was identified as P4 primase83 (**Supplementary Table 9**), detected within 58 species, which may suggest that P4 phage acts as a corridor of horizontal gene transfer between core members of the gut. A key driver of fungal evolution within gut ecosystems has been the transfer of functions from prokaryotes ⁸⁴, leading us to identify 87 protein clusters containing a eukaryotic protein that were also present in the prokaryotic reference genomes (**Figure 1g**). These findings support previous*

findings that horizontal gene transfer frequently occurs between members of the human gut microbiome ¹⁴.”

5.2. UHGP is built using a partially-overlapping set of source datasets which may explain why 3.3M proteins don't overlap with MiProtGut. However, the authors never discuss this significant set of proteins (~25% of UHGP-90) – can this be explained by different source data alone?

Response: As mentioned by the reviewer, the UHGP was focused on prokaryotes, so only used MAGs, rather than complete metagenomes. This means that even if the same samples had been studied originally, they would have ignored the majority of sequences as they would not fall into a MAG of sufficient quality for them to study. However, the reviewer highlights that the UHGP captured 3.3 million genes not observed within MiProGut. We believe these genes are due to the inclusion of genomes derived from isolates during the creation of UHGP. In our previous work we have isolated taxa from samples which could not be detected using metagenomics due to their low abundance (<https://doi.org/10.1016/j.chom.2022.09.011>). We have included this within the manuscript:

“The UHGP proteins not captured by MiProGut are likely due to the different sources each catalogue used to predict proteins, namely the focus on genomes for UHGP and metagenomes for MiProGut. As UHGP was generated from prokaryotic genomes, the majority of MiProGut proteins matching UHGP were of bacterial, archaeal, or unknown origin. Conversely, only 16.6% of viral and 0.3% of eukaryotic MiProGut proteins matched UHGP due to their exclusion (Supplementary Table 8).”

While this work represents the first iteration of MiProGut, we hope to improve it in future by inclusion of isolate genomes, both from our own work (<https://www.biorxiv.org/content/10.1101/2024.06.20.599854v1>) and others, allowing us to improve the catalogues diversity by including groups that are rarely captured by sequencing but are detectable by cultivation.

5.3. the MiProtGut collection contains a significant number of singletons (14M; line 300). It is not shown/discussed if singletons are shared between UHGP and MiProtGut or if they are unique. In the latter case, this increases the risk of the dataset being populated with false positives.

Response: While the initial number of singletons is daunting, we quote a recent larger-scale analysis of metagenomic data that independently showed a similar number of singletons occurring. To further support that these proteins are real, and not spurious, we studied the expression of these singletons, identifying that nearly 40% were observed to be expressed in at least one sample. This is highlighted with many of the small proteins shown in Figure 3d being singletons but were expressed in over 90% of the samples being studied. For these reasons we believe singletons can represent real proteins. We have expanded the discussion to highlight that while singletons exist, there is a rationale to including them:

“While many of the predicted proteins included within MiProGut were singletons, not clustering with any other predicted proteins, transcriptional evidence confirms the existence and that some are highly expressed, although rarely identified metagenomically. This may be due to them originating from low abundant members of the microbiota, which are often missed by sequencing⁹⁶. As gene catalogues are often

used as reference databases for the metagenomic study of an environment, the discovery of these missing proteins will facilitate the identification of disease-specific biomarkers that were previously overlooked.

6. I highly appreciate the use of metatranscriptomic data to further validate the predicted proteins. In this case, it would also be interesting to look at the taxonomic origin of proteins aligned to metatranscriptomic data to see if the increase in coverage is due to newly identified eukaryotic or prokaryotic proteins.

Response: This is a great suggestion which we hadn't considered. To address this we have now included the following section in the results detailing the number of proteins from these additional groups which were observed to be expressed:

“This included 22.7% of the protein clusters containing eukaryotic proteins, 35.9% of the viral containing clusters, and 45.4% of archaeal containing clusters.”

7. Small proteins are indeed a very overlooked group of proteins however, it is not without a reason. Many gene prediction tools reject short sequences to keep the false positive ratio low. Here authors modified the parameters of gene predictors to include such proteins. Unfortunately, this part is not so well described in the methods section. Furthermore, such a change might affect the false positive ratio of such methods. It would be beneficial to check how such change will affect the analysis or, at least, discuss it in the main text.

Response: Prior research has already shown that the human gut is a rich source of short proteins which appear to play important roles in the microbiota (Petruschke *et al*, 2020; Sberro *et al*, 2019). As only pyrodigal, based on prodigal, allowed for the predicted gene length to be altered we will focus on the rationale used in the original publication. In Hyatt *et al* (2010) they state “Although many of the short genes predicted by current programs that have no existing BLAST hits might be real, the likelihood is that most are false positives”. No evidence is provided to support this statement apart from a link to VerBerkmoes *et al* (2006) which conducted a proteomic study of a single strain. We observed that all predicted small proteins were detected as being expressed within at least one of the metatranscriptomic samples studied. We have now altered the results text to reflect this as such:

“Studying the expression of the SPCs within 687 metatranscriptomic samples revealed that all SPCs were expressed within at least one sample, but many were rarely expressed (<10% of samples). However, 69 SPCs were highly prevalent, being expressed in >90% of samples (Figure 3c).”

While further study is required to identify and apply solid threshold, we believe the transcriptional evidence we provide for the small proteins provides sufficient rationale for their inclusion and study.

8. The InvestiGUT is certainly a valuable tool, nevertheless it would benefit from extending the Readme section within the GitHub repository. Especially adding a short paragraph at the beginning describing the purpose of the tool could be helpful to users who did not read the text carefully.

Response: Expanding the GitHub is a wonderful idea to highlight the usefulness of InvestiGUT to users who may not have read the paper. We have now included a “What

is InvestIGUT” and “How does it work” section to provide users with a reminder. Once published we will include a link to the manuscript here as well to direct readers to further information.

9. In Figures 3 and 4 the fonts are tiny and difficult to read. I am open to the authors making the figures more legible in whichever way they see fit. My suggestion for Figure 3 would be to parse names in panel D to make them shorter and easier to read. For Figure 4, I would suggest splitting it into 2 figures – panel A as a figure, then panels B-F a separate figure.

Response: We understand the reviewers' concerns about the text size in both of these figures and agree they are quite small. We have adapted the figures to attempt to make them more legible.

Figure 3 has had most text increased by 30% to enhance readability, however the protein names remain an issue. We believe this must remain part of the main text due to the importance of linking these proteins to the text.

Figure 4 has been split into two separate figures as suggested by the reviewer, allowing the figures to be increased in size and given more space to separate and be understood.

Some more minor comments about Figs. 3 and 4:

• **Fig. 3A:** top bar shows length 0-50. In fact, according to the Methods section the minimum length is 6 residues

• **Fig. 3C:** there are no units given for the Y-axis (should it be %?)

• **Fig. 4E:** There are no studies identified as data sources. Are those the same studies as in panel (f)?

• **Fig. 4 caption:** The caption is incompatible with previous captions. For example, it moves defining the panel (a), (b), etc. to ends of sentences. Also, it describes some results in full sentences, which is fine, but also inconsistent with previous captions.

Response: The figures have been changed as suggested by the reviewers, as have the captions.

Minor comments

• **abstract:** The MiProtGut collection is not mentioned in the abstract at all – I would welcome including this important mention in the abstract

Response: MiProGut is now named within the abstract.

• **I. 68:** has -> was

Response: Changed.

• **I. 70:** other taxa -> other domains (or clades?)

Response: Changed to domains.

• **I. 87-90:** explanation of why Pyrodigal was used is phrased in a confusing way.

It would be sufficient to say that Pyrodigal and Prodigal give identical results (l. 88-89), hence only one was used.

Response: We agree and have changed the text to simply the rationale as suggested.

• **l. 99-101: the authors evaluate 26 species from different domains of life but no Eukaryotes. Why? Is there a specific reason for that?**

Response: We think there is some confusion about this as we included three fungal species in the analysis, namely those of particular interest to the gut. To clarify this we have added "eukaryotes/fungal".

• **l. 128-130: in lines 111-113 the authors mention removing the human genome data using BMap, then in lines 128-130 they do it again, but using Kraken2. Was that necessary? If yes, I welcome a comment on why it was necessary and some data on how much improvement did the second pass-through using Kraken2 provided**

Response: BMap was used to filter the raw reads for samples we assembled ourselves, while Kraken2 was used mainly to filter those assembled metagenomes we obtained from community resources. We do not believe such a "two pass" system is needed for day to day analysis.

• **l. 148: six amino acids plus stop codon -> six amino acids plus a stop codon**

Response: Changed.

• **l. 154: Convention suggests that upon release v. 1.0 is used. Why InvestiGUT is still in v. 0.1?**

Response: We wanted to wait until publication until committing to version 1, but have now released v. 1.0 after incorporating the reviewers suggestions for edits to the code and README.

• **l. 154-164: This paragraph includes a very technical description of the InvestiGUT package. Most of it could be moved to the README section of the repository without detriment to the publication.**

Response: We believe this information is still key to the publication so it can stand-alone, but we agree that this text is useful in the README as it provides needed information. Due to this we have extended the GitHub README to include this information.

• **l. 192-201: the paragraph discusses functional annotations but there is no mention of the ontology that is being produced. Is GO or KEGG used? Or are those just headers from a protein database (which?)**

Response: As mentioned, we used eggNOG mapper, which uses the COG classification system, hence it is used in Figure 2b where the functionality is studied. Apart from eggNOG, MANTIS applies a merger of many databases and ontologies while DBAASP does not provide ontology but predicted activity against strains of medical interest.

• **Figure 1A: dereplicated suggests removing redundancy (clustering at 100% sequence identity), while the number of genes termed „dereplicated” corresponds to the MiProtGut-90 (clustering at 90% seq. id.)**

Response: We agree and have changed it to state "Protein clusters at 90% similarity".

- **I. 207-210: equations in their current form are hard to read, especially the equation in line 210**

Response: We understand the reviewers comments but have been unable to come up with an alternative solution.

- **Figure 2B: The scale on the Y-axis is missing. As is, it reads that the scale goes up to 3 protein clusters. Also, there is a break between the end of the grey color bar and 3.0 – what is the actual maximum?**

Response: We thank the reviewers for noticing this, it is at 10 to the power of 7. The break is intentional as we have 29.2 millions protein clusters. To clarify this, we now state the exact number in the figure legend.

- **I. 275: refraction -> rarefaction**

Response: Changed.

- **I. 276: functional annotation shown here is from eggNOG-mapper, MANTIS, DBAASP, or a sum of all?**

Response: We have clarified in the text this is from eggNOG-mapper alone.

- **I. 292: permeations -> permutations (?)**

Response: Changed.

- **I. 319: it's unclear to me why the number of clusters is expressed as integers and not natural numbers**

Response: We realized that the way we discuss these numbers is quite confusing. These numbers account for when a protein cluster is assigned to multiple COG categories, meaning whole numbers are not always possible. To clarify this we now refer to the number of protein clusters that are assigned to that functional category, either solely or in addition to other functions. This has increased the number of proteins assigned to 'nuclear structure' in MiProGut to 413, while UHGP has remained much lower, at only 7 protein clusters.

- **I. 322-323: should this be the final sentence of the previous paragraph? Also, there is a connecting sentence missing linking the analysis of MiProtGut-UHGP overlap to the analysis of metatranscriptomic data**

Response: We agree the sentence was misplaced and have also included a further two sentences to better allow the transition between sections.

- **I. 324: Figure 2c -> Figure 2d**

Response: Changed.

- **I. 392-393: phrase „where only samples that are positive for all queried sequences are reported” was unclear to me until I reached the subsequent paragraph. Please, consider rephrasing.**

Response: We appreciate the consideration of clarity and have changed the text in accordance.

- **I. 404: who are the 4,464 individuals? The phrasing is unclear whether this is only the case in the particular analysis the authors describe, or if it is a feature**

of InvestiGUT that it always performs a comparison against an arbitrary set of individuals.

Response: This sentence has been expanded to clarify it is the individuals gut microbiome which is being studied and that they are a defined cohort from the Netherlands and Israel which are studied.

• I. 415: gender -> sex

Response: Changed throughout the manuscript.

• I. 437: are the 69 SPC the same as in Figure 3C? Maybe reference?

Response: Yes they are, we now refer back to the previous figure to clarify this.

• I. 440 & 447: encoding of protein identifiers is never explained in the text and hard to read in its current format

Response: Thanks a lot for pointing out this error, we have now included a section in the methods which details the system we designing for naming the proteins. We agree that they are hard to read, but they do contain all the relevant information needed to understand the proteins origin.

Reviewer #1 (Remarks on code availability):
I checked the repository but didn't run the code. Some comments about the code are included in the comments to the authors.

Response: According to the reviewers comments we have improved the GitHub.

Reviewer #2 (Remarks to the Author):

Response: We thank the reviewer for taking the time to study our manuscript and contribute to improving the work.

Reviewer #3 (Remarks to the Author):

The manuscript "Lineage-specific microbial protein prediction enables large-scale exploration of protein ecology within the human gut" by Schmitz et al. describes an important issue in the field of metagenomics, the problem that most metagenome workflows rely on general purpose gene prediction instead of incorporating a lineage-specific ORF prediction on assembled contigs. They present a new method, InvestiGUT, that enables the use of the correct genetic code based on taxonomic assignment of genomic fragments, removes partial predictions and incorporates the prediction of small proteins. The tool itself is easy to install and runs the given example correctly after following the installation instructions.

Response: We appreciate the time you have taken to provide feedback for our project. Also, thank you for taking the time to test the code and we are glad you were able to get it running without issue.

We feel that the manuscript has some major issues that need further revision:
- The introduction and discussion lack on focus on the proposed contribution: Why are we struggling to predict correct ORFs ? What could be done instead ? Why this is an overall problem? Can we quantify/estimate the problem of incorrectly predicted genes in public databases or MAG/gene catalogs? What are the current approaches to overcome this or why lineage-specific prediction is a necessary approach that needs to be implemented

Response: We agree that highlighting these issues further within the introduction will help the reader understand why lineage specific gene prediction methods are needed, and hence the reason for this work. We have now added a paragraph to the introduction outlining these issues:

“Further to this, several critical issues in genome annotation, particularly when applied across diverse taxa have been identified²⁹. One of the major challenges identified is the variability in gene prediction accuracy, where certain tools excel at predicting specific gene types in particular taxa, but fail when applied to others. This inconsistency stems from the vast diversity in genetic structures, including variations in coding sequences, regulatory elements, and the genetic codes used by different organisms. Additionally, prokaryotic annotation tools often perform poorly when applied to eukaryotic genes, especially those with complex exon-intron structures, while tools designed for eukaryotic genomes may overlook small, overlapping genes commonly found in prokaryotes³⁰. The limitations of current annotation tools are compounded by the lack of comprehensive training datasets, particularly for non-model organisms, especially lacking those from diverse lineages, leading to errors in gene predictions and functional annotations. As a result, the incomplete or inaccurate annotation of metagenomic datasets can obscure important biological functions, particularly in diverse microbial communities. These challenges have started to be addressed by the introduction of ‘re-annotation’ techniques³¹ and initiatives to clean-up consensus assemblies and annotations^{32,33}. While these solutions address genomes, microbiota vary greatly, meaning de novo gene prediction is essential. As of yet no lineage directed gene prediction method exist for microbiome analysis.”

- In order to prove its usefulness for metagenomic applications, benchmarking of the ensemble gene calling approach should include more organisms (if well-annotated representatives are available) covering a wide variety of genetic codes and/or species with peculiarities, e.g. alternative start codons (e.g. Actinobacteria).

Response: While we can understand the reviewers comment about benchmarking more organisms, this is not the focus of the study. We have added more background and reasoning as to why we selected these genomes to benchmark in this study. Briefly, prior benchmarking was recently done in the ‘ORForise’ study, identifying substantial variation in gene prediction when using 6 bacterial genomes from 2 phyla. We have expanded this to 26 species, including additional Bacteria but also expanding the analysis to include Archaea, Viruses, and Eukaryotes. We believe a more directed approach of genome inclusion serves as a better selection than simply adding more genomes with the same issues already identified. This is why we focused on microbes found in the gut. We do not claim the included benchmarking will apply for other ecosystems, but instead each ecosystem needs its own benchmarking depending on

the taxa present. To highlight this in the paper we have included additional text to the discussion:

*“We also observed ~10% variation in the ability of gene prediction tools to correctly predict genes on both fungal and viral genomes. This reinforces the need to select a tool based on the lineage of the sequence being studied. While the variation in perfect predictions between tools was large, the variation was much lower for partial predictions. Interestingly we observed that all tools performed poorly on *Aspergillus nidulans*, with single tools predicting less than 9% of the genes, while a combination of three increased the perfect predictions to 14%. When applied to the human gut, this approach uncovered eukaryotic, archaeal and viral proteins that have previously been overlooked²⁷. While the application of multiple gene prediction tools increased the prediction of correct genes, it also increased the number of spurious predictions. As these proteins will not match queried proteins in InvestiGUT or match data aligned to MiProGut this does not pose an issue, but highlights that further improvements are needed to correctly identify genes from microbiomes. As the selection of gene prediction tools was selected based on the benchmarking on gut specific species, further benchmarking is needed before application to additional ecosystems.”*

**- Most importantly, the ensemble ORF prediction workflow, which is currently not part of the git repository should be available as tool/pipeline:
o Where is the prediction workflow code?**

Response: We have now included the ORF prediction workflow on the GitHub as a branch of the repository called ‘nextflow-testing’ to allow users to apply the same prediction workflow themselves. This is clarified in the text:

“InvestiGUT, along with the protein prediction pipeline used in this work is available at: <https://github.com/Matt-Schmitz/InvestiGut>.”

If the reviewers are instead referring not to our own code, but the benchmarking which was done using the published ORForise tool by Dimonaco et al (2022) and is available for others to apply.

Minor points:

- There should be functionality to build your own catalogue.

Response: This is beyond the focus on this manuscript. We do not intend for new catalogues to be created by each user as both the prediction and catalogue creations requires quite a lot of resources, hence we provide all the precomputed proteins. However, we do now also provide each samples individual proteins, as well as multiple additional clustering levels to enhance the usability of the dataset.

- The reason for the sequence identity thresholds chosen for clustering during catalogue creation should have a function/mechanism-driven explanation, rather than just being the ones used in UHGP-90.

Response: We believed this allowed for direct comparability to a major resource used by the community, but do agree. To amend this we also provide additional clustering levels for the proteins, including the raw proteins unclustered, and dereplicated at 100% similarity.

- Parameters for eggNOG-mapper/MANTIS are missing (or please mention if all are default).

Response: We now confirm in the methods that default parameters were used when applying both tools.

- In the Supplement for the prediction benchmark, please include a column with the percentage of spurious predictions?

Response: We initially did not include the percentage for spurious predictions as the number can be infinite, and is not constrained by the number of true predictions. Supplementary Table 3 has been modified to include an additional column which is the percentage of spurious predictions.

- Regarding the ORF prediction benchmark results from ORForizer and merging 1, 2, and 3 tool combinations:

- For all tools, relying on a single tool with the 'best' results (lowest spurious rate) seems to lead to much more favorable real ORF to spurious ratios.

- Even when using three tools, the 'best' combination in terms of minimizing spurious predictions was not chosen for catalog generation. Instead, the combination that gave the most predictions was selected, even though the benchmark data showed an extreme inflation of spurious predictions compared to the tool with the lowest spurious ratio. This observation is based on the modified and merged results tables per domain in '515720_0_supp_652116_sdxq81_review_mod.ods'. It is critical to clarify the reasons for this choice, as it could have a significant impact on the quality and reliability of the resulting catalog.

Response: This is correct, as our goal was to facilitate the study of a protein's ecology within the gut, we believed it was more important to improve our 'true' prediction rate and have additional spurious proteins predicted, than miss some of the real proteins. This is because when analysing the data the spurious proteins would be ignored. Since having built the protein catalogue as a secondary outcome of the work, the same factor is true, either data aligns to the protein catalog, validating its existence and use to the researcher, or it doesn't and the protein can be ignored.

We agree with the reviewers that this rationale wasn't clearly stated within the manuscript, so have now edited both the results and discussion to clearly state that while no method is perfect, we have enhanced our ability to study functional protein ecology, but more refinement in the methods are required. As this is a focus of the groups research moving forward we appreciate the reviewers comments on current limitations and suggestions as we do aim to examine each of these aspects in future.

In the results we provide this frank analysis of the results:

“While this approach does result in an inflation of spurious predictions, below we have applied multiple approaches to study the benefit of this approach. This includes metatranscriptomic analysis to identify evidence for expression, and comparison with an independent gene catalogue to provide independent validation. Based on these results, we believe it is more advantageous to predict additional real proteins at the expense of including spurious genes, than to risk exclude real proteins.”

In the discussion we have now included the following test:

“As these proteins will not match queried proteins in InvestiGUT or match data aligned to MiProGut this does not pose an issue, but highlights that further improvements are needed to correctly identify genes from microbiomes.”

- Inconsistency of tools used across domains in Figure 1: There appears to be a mismatch between the tools used for each domain in section (a) and the tools compared per domain in section (d). Specifically, it seems that the tool sets for viruses and archaea may have been mixed up.

Response: Thank you for noticing this error. You are correct that the Viruses and Archaea tools were swapped in Figure 1a and has now been corrected. We have now double checked the numbers as well to ensure no further mistakes have occurred due to this.

- The overlap with UHGP-90 doesn't prove non-spuriousness of predictions since it was created using prodigal and cannot be claimed to be error free either.

Response: We agree with the reviewer that matching with a prior gene catalogue does not disprove they are spurious, although does provide support that they have been identified before, similar to the metric of “can we find a similar gene in NCBI-NR” which has been previously used. We have adjusted the text in the paper to reflect this and tone down our response to these matches:

“While the identification of overlapping proteins between the two catalogues supports that these proteins have been observed previously, further evidence is required to validate them as real and not spurious.”

- The use of metatranscriptomics data as a control is a good approach. It could be used to filter out predicted ORFs with no metaT support, which are then likely to be spurious (and perhaps could also be used to calibrate the ensemble ORF prediction approach on real-world data).

Response: While we agree that filtering for expression could be done using the metatranscriptomic data, we have metagenomes for thousands of patients, and metatranscriptomes for few fewer. Due to this, we lack any metatranscriptomic data from many continents, preventing us from really saying if geographically unique proteins are real or not. For what the reviewer proposes with calibration of ORF prediction we again agree that this would be a wonderful approach, but require paired data from a large enough cohort. In future we would like to pursue this to improve InvestiGUT and gene prediction as a whole.

- The small protein and AMP parts of the manuscript are a nice side tangent of the project but feel slightly disconnected with the ‘main’ part, i.e. the prediction workflow and its validation (and subsequently the validity of the catalog).

Response: We believe the prediction of small proteins is one of the main improvements of our approach compared to those used by most of the community. We use the presence of AMPs to highlight that not only are these small proteins real (as confirmed by metatranscriptomics) but of real use to the survival of the strains producing them via an understandable mechanism i.e. killing others. To make this

rationale clearer within the manuscript we have edited the results section introducing the AMPs.

- Some suggestions to make the tool more user-friendly and runnable on large-scale datasets

o Allow users to control the number of threads for DIAMOND search

o Block size to optimize search time

o E-value for catalog search is hardcoded to 1e-3; make it user-adjustable?

Response: We can see the benefit of such as option, to facilitate this, but account for the vast number of options DIAMOND can accept, we have modified the code to shunt all unrecognised command line options to DIAMOND. This ensures changes in DIAMOND will also be accounted for automatically within InvestiGUT.

Reviewer #3 (Remarks on code availability):

See main review, the installation and the example is working, the code for the prediction is missing.

Response: We thank the reviewer for checking our code. You are correct that the code for prediction has not been made available. Our initial goal was to create the InvestiGUT tool, not a gene prediction pipeline. However, given these comments we have included the pipeline into the manuscript and included it within the GitHub repository.

Reviewer #4 (Remarks to the Author):

Response: We thank the reviewer for taking the time to study our manuscript and contribute to improving the work.

Response to reviewers

Reviewer #1 (Remarks to the Author):

I want to express my gratitude to the authors for their meticulous attention to the issues raised in my feedback. All of the minor comments have been satisfactorily addressed. Major comments, in principle, are also well-addressed.

The revision process has also helped me gain a deeper understanding of the Authors' intentions behind this work. Considering that the primary research outcome of this work is the InvestiGUT, I would like to suggest that the Authors consider how the MiProGut collection is presented and discussed. In particular, MiProGut is likely to be perceived as an alternative to the UHGP collection and be used also outside the InvestiGUT framework. While UHGP may be incomplete, some effort by the authors of UHGP work was put into minimizing the number of false positives. Since MiProGut relies on a sum of different predictions, it is possible that it may have more false positive genes, even though the Authors provide some reassurance that this may not be a significant issue through the metatranscriptomics case-study.

Response: We agree with the reviewer that the use of multiple different prediction tools will lead to increased false positive gene predictions and have highlighted this in the discussion. The reviewer's suggestion for a high-quality filtered version of MiProGut has allowed us to alleviate any concerns as 86.4% of protein clusters were retained.

To address this, I propose some alternative solutions:

1. Provide a broader discussion about the nature and limitations of MiProGut in the paper (ideally also by including appropriate notice in READMEs). For instance, you could expand upon the existing discussion in lines 562-581.

Response: Given the reviewers suggestion for creating a high-quality sub-set of MiProGut, we have expanded the discussion to highlight why we have done this, and why the criteria for filtering were selected. This has substantially expanded this section of the discussion from line 565 to 597.

2. Create a MiProGut-HQ (high-quality) collection by intersecting gene predictors or employing a different "false positive" detection method. This approach would make the collection more appealing to researchers studying microbial proteins in other non-omics contexts where the issue of false positives could be particularly significant.

Response: A high-quality version of MiProGut makes a lot of sense. We used the expression of a protein across the studied metatranscriptomic samples, prediction of a protein at least twice, and the prediction of a protein using multiple tools to identify high-quality predictions. This provided a final set of 25,266,245 proteins (86.4%). MiProGut-HQ is detailed in the results at line 389 - 392 and the discussion at line 584 - 595.

Reviewer #1 (Remarks on code availability): I did code assessment during my first round of reviews. I did not run the code this time around, but I did study the changes made by the Authors since the last review.

Response: Thank you for taking the time to look over our edits.

Reviewer #3 (Remarks to the Author): The authors have responded sufficiently to all our comments. Only problems with the git, see below Thanks Reviewer #3 (Remarks on code availability): Tested on Macbook Apple M2 Pro, 16 GB, Sequoia 15.2, with the given command

```
mamba create --no-channel-priority -n investigut > -c bioconda -c conda-forge >
"python=3.11" "numpy=1.24.3" "scipy=1.10.1" > "conda-forge::matplotlib-base"
"seaborn=0.13.0" > "pandas=1.5.3" "statsmodels=0.13.5" "ete3=3.1.2" >
"openpyxl=3.0.10" "bioconda::diamond=2.1.8"
```

I get the following error:

error libmamba Could not solve for environment specs

The following package could not be installed

└─ diamond =2.1.8 * does not exist (perhaps a typo or a missing channel).

Response: Thank you for checking our code and trying to install it on your system. It turns out the version of diamond we proposed was only for Linux systems and not Mac, we have now altered the code to rely on a different version of DIAMOND that is available on both systems, so should be installable. An update to the GitHub was made to solve this issue.

Reviewer #4 (Remarks to the Author): I co-reviewed this manuscript with one of the reviewers who provided the listed reports. This is part of the Nature Communications initiative to facilitate training in peer review and to provide appropriate recognition for Early Career Researchers who co-review manuscripts.

Response: We appreciate your effort in reviewing our work.